# EQUILIBRIUM LANGUAGE MODELS

**Yikun Jiang**[1], **Huanyu Wang**[1], **Tianhong Ding**[1],
**Wenhu Zhang**[2], **Yiming Wu**[3], **Hanbin Zhao**[4],[*] **John C.S. Lui**[5]
[1]Huawei Technologies Co., Ltd., [2]The Hong Kong University of Science and Technology,
[3]The University of Hong Kong, [4]Zhejiang University, [5]Chinese University of Hong Kong
jiang_yikun@pku.edu.cn

## ABSTRACT

Large Language Models (LLMs) excel across diverse applications but remain impractical for edge deployment due to severe memory bottlenecks at the edge devices. We propose Equilibrium Language Models (ELMs), a novel compression framework that replaces groups of Transformer layers with a lightweight fixed-point network, reinterpreting deep computation as solving for an equilibrium state. To achieve ELMs, We introduce *Group Pruning Policy Optimization*, which automatically learns optimal pruning intervals. Moreover, we propose *One-Step KV-Cache*, which drastically reduces memory overhead by storing only the final iteration cache without compromising the accuracy, to enable effective deployment at the edge devices. Across different tasks such as common sense reasoning, mathematical problem solving, and code generation, ELMs prune 28% of parameters while retaining 99% of the accuracy of dense fine-tuned LLMs, establishing a new direction for memory-efficient edge deployment of large models. We provide the implementation of our method in https://github.com/Jyk-122/ELM.

## 1 INTRODUCTION

Large language models (LLMs) have demonstrated remarkable success in diverse applications, including dialogue systems, code generation, and mathematical reasoning. As the demand for mobile and edge intelligence continues to grow, on-device deployment of LLMs has become an inevitable trend. However, such deployment is fundamentally constrained by the limited capacity of edge devices, where storing billions of parameters in RAM at inference time often exceeds the available memory budget. This resource bottleneck has spurred extensive research into model compression techniques, such as pruningAshkboos et al. (2024); Xia et al. (2023); Frantar & Alistarh (2023); Sun et al. (2023); Ma et al. (2023); Lu et al. (2024), quantization (Frantar et al., 2022; Xiao et al., 2023; Lin et al., 2024), and knowledge distillation (Hinton et al., 2015; Xu et al., 2024; Sreenivas et al., 2024). Overall, common compression methods can be categorized into two groups, *i.e.*, (1) simultaneously reducing both computational cost and model size, and (2) reducing computational cost only.

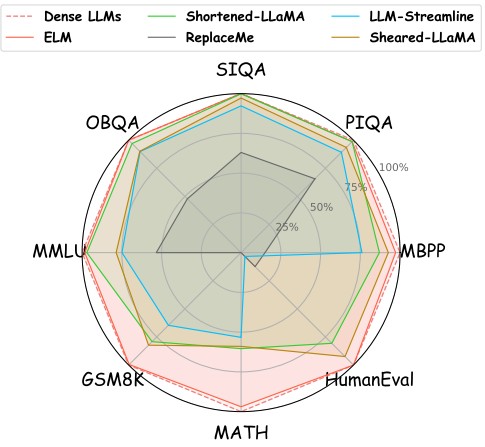

Figure 1: Benchmark evaluation of pruning baselines on Qwen2.5-7B-Instruct.

Methods in the first category typically leverage statistical metrics to prune low-contribution parameters (Men et al., 2024; Kim et al., 2024; Yang et al., 2024), or replace consecutive intermediate layers with lightweight modules (Chen et al., 2024; Shopkhoev et al., 2025). While effective in reducing model size, these approaches often degrade performance on challenging generative tasks, such as

---

[*]Corresponding Author.

code synthesis and mathematical problem solving, due to the loss of model capacity. Methods in the second category adaptively skip layers conditioned on the input (Raposo et al., 2024; Jiang et al., 2024; Tan et al., 2024), thereby reducing computation at inference, yet they offer minimal savings in parameter and fail to meet the stringent low-memory requirements of edge devices. Compared with these approaches, our method focuses on alleviating the memory bottleneck of edge deployment, enabling LLMs to run efficiently on constrained devices without compromising accuracy.

To this end, we propose **Equilibrium Language Models (ELMs)**, a novel model compression framework for Transformer-based LLMs. Rather than pruning individual network components, which often results in significant accuracy loss, ELMs replace groups of intermediate transformer layers with a lightweight fixed-point network. In this formulation, the computation of a deep layer stack is reinterpreted as solving for an *equilibrium state* of a fixed-point system conditioned on the input. Formally, the output is defined as the solution $z^*$ to a non-linear system, that is, $z^* = f(z^*, x)$, where $x$ represents the conditional input. Drawing on theoretical insights from deep equilibrium models (Bai et al., 2019), this implicit formulation achieves a representational capacity comparable to explicit deep stacks, while requiring significantly fewer parameters.

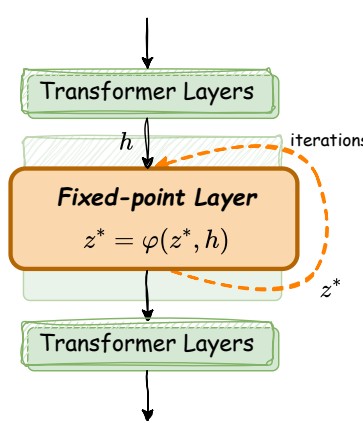

Figure 2: Illustration of ELMs. The group of intermediate transformer layers is replaced with a lightweight fixed-point layer.

Implementing ELMs requires addressing two critical challenges: *determining which layer interval to compress as the fixed-point network* and *how to efficiently handle the KV cache during inference*. Prior methods typically rely on heuristic metrics such as cosine similarity or perplexity to determine the importance of each layer. However, such metric-based criteria cannot be directly applied to ELMs, since they fail to capture the approximation error that arises when substituting multiple layers with a fixed-point formulation. To address this limitation, we introduce *Group Pruning Policy Optimization (GPPO)*, a reinforcement learning policy optimization(Sutton et al., 1999) framework that automatically identifies the optimal layer indices for equilibrium modeling. Concretely, GPPO samples the index of the first candidate layer from a learnable distribution, and utilizes a supervised fine-tuning loss as the reward signal. In addition, we propose *One-Step KV*, which retains only the cache from the final iteration during inference. This approach dramatically reduces memory overhead comparing with storing caches for all intermediate iterations, without compromising accuracy. Moreover, it enables token-adaptive fixed-point iterations to proceed without recomputing missing intermediate KV caches.

We evaluate ELMs on three challenging downstream tasks: common sense reasoning, mathematical problem solving, and code generation. Our method consistently outperforms the state-of-the-art methods. Even with a 28% reduction in parameters, ELMs retain on average 99% of the accuracy of fully fine-tuned dense LLMs, demonstrating the effectiveness of deploying on edge devices.

## 2 PRELIMINARY

We next review autoregressive language models with KV caching and fixed-point networks with equilibrium-based training, which form the foundation of our method.

### 2.1 AUTOREGRESSIVE LANGUAGE MODELS

Large language models are typically built upon decoder-only Transformer architectures (Vaswani et al., 2017), where tokens are generated in an autoregressive manner. Each Transformer layer employs a residual structure that integrates multi-head self-attention with a feed-forward network. Formally, the operation of the $l$-th layer can be expressed as

$$\boldsymbol{h}^{l+1} = f^l(\boldsymbol{h}^l) = \text{FFN}(\text{MHA}(\boldsymbol{h}^l)), \tag{1}$$

where $\boldsymbol{h}^l$ denotes the hidden state input to the $l$-th layer, MHA represents the multi-head attention module and FFN is the feed-forward network. In decoder-only Transformers, a causal mask is

applied to prevent information from subsequent tokens from leaking into preceding ones. This property enables caching of key and value tensors in self-attention (Pope et al., 2023), thereby avoiding redundant computations. For instance, when generating the $(T + 1)$-th token, the output of the $l$-th layer can be written as

$$\boldsymbol{h}_{T+1}^{l+1} = f(\boldsymbol{K}_{1\cdots T}^{l}, \boldsymbol{V}_{1\cdots T}^{l}, \boldsymbol{h}_{T+1}^{l}), \tag{2}$$

where $\boldsymbol{K}_{1\cdots T}^{l}$ and $\boldsymbol{V}_{1\cdots T}^{l}$ represent the cached key and value tensors of preceding tokens at the $l$-th layer. This KV cache mechanism has become a standard acceleration technique for LLM inference, reducing the attention complexity from $\mathcal{O}(T^2)$ to $\mathcal{O}(T)$.

## 2.2 FIXED-POINT NETWORKS

Conventional neural networks transform features by explicitly computing forward through a stack of layers. In contrast, fixed-point networks, a class of implicit neural networks, define the model output as the equilibrium solution of the fixed-point equation $z^* = f(z^*, x)$. The equilibrium state is obtained through an iterative optimization process,

$$z^* = \lim_{k \to \infty} z^{(k+1)} = \lim_{k \to \infty} f(z^{(k)}, x). \tag{3}$$

The superscript $k$ denotes the iteration step. Formally, fixed-point network is equivalent to a weight-tied network of infinite depth, thereby achieving stronger expressivity with fewer parameters. However, solving for the fixed-point generally requires a variable number of iterations and incurs substantial memory overhead during training. Deep equilibrium models (DEQs) address this challenge by analytically backpropagating through the equilibrium point $z^*$ using implicit differentiation, which eliminates the need to store intermediate activations and results in constant memory usage:

$$\frac{\partial z^*(\cdot)}{\partial(\cdot)} = (I - \frac{\partial f(z^*, x)}{\partial z^*})^{-1} \frac{\partial f(z^*, x)}{\partial(\cdot)}. \tag{4}$$

To accelerate training, Jacobian-Free Backpropagation (JFB) (Fung et al., 2022) drops the inverse Jacobian term and instead computes the gradient only at the final iteration, thereby circumventing the cost of matrix inversion. Stochastic JFB (SJFB)(Bai & Melas-Kyriazi, 2024) introduces randomized iteration steps during training, achieving superior results in large-scale experiments.

## 3 EQUILIBRIUM LANGUAGE MODELS

### 3.1 WORKFLOW OVERVIEW

From a holistic perspective, an Equilibrium Language Model (ELM) replaces consecutive redundant layers as a fixed-point module in the transformer architecture. Given a language model $\mathcal{F}$, which contains $N$ transformer layers, we prune consecutive $M(M < N)$ layers

$$\mathcal{F}(x) \equiv f^{N-1} \circ \cdots \circ \underbrace{f^{l+M-1} \circ \cdots \circ f^l}_{\text{Pruned Layers}} \circ \cdots \circ f^0(x). \tag{5}$$

Following this pruning procedure, we embed a fixed-point module $f^*$ at the position of the $l$-th transformer layer and the model becomes

$$\hat{\mathcal{F}}(x) \equiv f^{N-1} \circ \cdots \circ \underbrace{\varphi}_{\text{Fixed-Point Module}} \circ \cdots \circ f^0(x). \tag{6}$$

Let $\boldsymbol{h}^l \in \mathbb{R}^d$ denote the conditional input of the fixed point module and $d$ represents the dimension of hidden feature. We denote the iterative value as $\boldsymbol{z}^{(k)} \in \mathbb{R}^{d \times d}$, where $k$ represents the iteration step. The module $\varphi$ produces the equilibrium state $\boldsymbol{z}^* = \varphi(\boldsymbol{z}^*, \boldsymbol{h}^l)$ through iterative rule

$$\boldsymbol{z}^* = \lim_{k \to \infty} \boldsymbol{z}^{(k+1)} = \lim_{k \to \infty} \varphi(\boldsymbol{z}^{(k)}, \boldsymbol{h}^l). \tag{7}$$

The implicit fixed-point layer to be trained is composed of two fully connected layers $\boldsymbol{W_z}, \boldsymbol{W_h} \in \mathbb{R}^{d \times d}$ and a transformer layer $f^*$. The network of the fixed-point module is defined as

$$\varphi(\boldsymbol{z}^{(k)}, \boldsymbol{h}^l) = f^*(\boldsymbol{W_z} \cdot \boldsymbol{z}^{(k)} + \boldsymbol{W_h} \cdot \boldsymbol{h}^l). \tag{8}$$

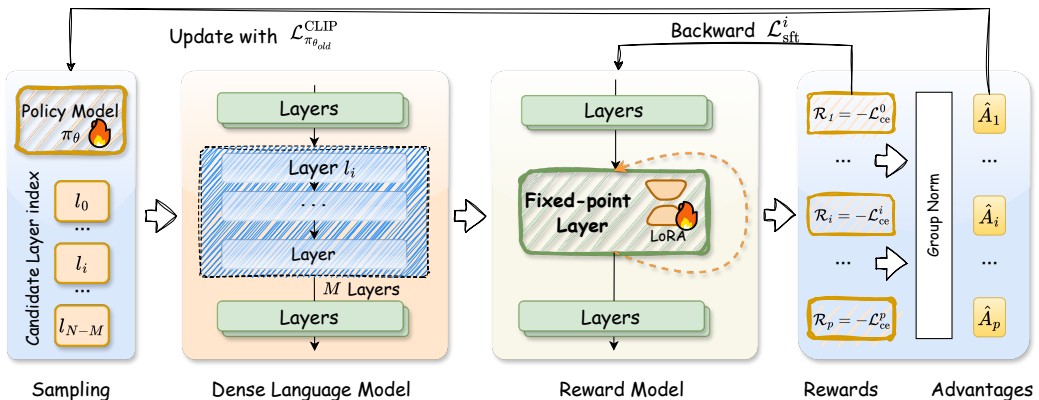

Figure 3: Workflow of GPPO. The policy model samples groups of pruned layers, which are used to construct ELM as reward models. Each fixed-point layer is trained with LoRA, updated by supervised loss, while group-normalized rewards guide the policy update.

The transformer layer $f^*$ inherits parameters from the first pruned layer $f^l$, and the initial value of $\boldsymbol{z}^{(0)}$ is set to zero. In addition, the learnable matrix $\boldsymbol{W_z}$ is random initialized and $\boldsymbol{W_h}$ is set as an identity matrix. Consequently, the result of fixed-point layer $\varphi$ at the first iteration is the same as the output of layer $f^l$, enabling a good initial starting point.

Although the upper bound of iteration numbers is $+\infty$ in theory, the iteration converges within $M$ steps in empirical practice. In this way, the proposed ELM achieves parameter compression over the $M$ transformer layers, while incurring no additional computational complexity.

The optimization objective of the proposed fixed-point module is composed of two components: a cross-entropy loss and a distillation loss, as

$$
\begin{aligned}
\mathcal{L}_{\text{sft}} &= \lambda_{\text{ce}}\mathcal{L}_{\text{ce}} \qquad\quad +\lambda_{\text{distill}}\mathcal{L}_{\text{distill}} \\
&= \lambda_{\text{ce}}\mathbb{E}_{\mathcal{D}}\text{CE}(\hat{\mathcal{F}})+\lambda_{\text{distill}}\mathbb{E}_{\mathcal{D}}\text{MSE}(\boldsymbol{h}^{l+M}, \boldsymbol{z}^*)
\end{aligned}
\tag{9}
$$

where $\mathcal{D}$ denotes data samples from the training dataset. $\lambda_{\text{ce}}, \lambda_{\text{distill}}$ are hyper-parameters of the cross entropy and distillation respectively. We utilize a distillation loss (MSE) to align the output of $\varphi$ with the output hidden states $\boldsymbol{h}^{l+M}$ from the $(l+M-1)$-th layer $f^{l+M-1}$ of baseline transformer $\mathcal{F}$. Additionally, a cross-entropy loss is employed to optimize the next-token prediction results.

Training fixed-point networks with implicit gradient in Eq. 4 is computationally expensive due to matrix inversion. To mitigate this, we adopt the Stochastic Jacobian-Free Backpropagation method for training the fixed-point layer. Specifically, we uniformly sample $m \sim U[0, M/2]$ steps of fixed-point iterations without gradients and $n \sim U[1, M/2]$ steps with gradients, as illustrated in A.1.

## 3.2 GROUP PRUNING POLICY OPTIMIZATION

Inspired by the success of policy optimization methods such as PPO (Schulman et al., 2017) and GRPO (Shao et al., 2024), we develop Group Pruning Policy Optimization (GPPO), a lightweight framework for automatically search for pruning intervals under the ELM formulation. GPPO follows the policy gradient paradigm, where learnable parameters $\theta \in \mathbb{R}^{N-M+1}$ define a policy model that represents the probability of selecting each candidate layer as the first pruned layer. The resulting compressed equilibrium model serves as the reward model, with the policy distribution $\pi_\theta$ adapting to maximize downstream performance as shown in Fig. 3.

Concretely, each transformer layer $f^0, \ldots, f^{N-M}$ is treated as a candidate, and a lightweight LoRA module is attached to reduce training overhead. A layer index is sampled from $\pi_\theta$, and the supervised fine-tuning loss $\mathcal{L}_{\text{sft}}$ (Eq. 9) is computed to update the reward model. At the same time, negative cross-entropy losses are accumulated as rewards,

$$
\mathcal{R} = \{-\mathcal{L}_{\text{ce}}^1, \cdots, -\mathcal{L}_{\text{ce}}^i, \cdots, -\mathcal{L}_{\text{ce}}^p\},
\tag{10}
$$

---

**Algorithm 1** Group Pruning Policy Optimization (GPPO)

---

**Require:** Policy parameters $\theta$, Original language model $\mathcal{F}$,
**Hyperparameters:** Clipping threshold $\epsilon$, Policy update frequency $p$, Policy update iterations $q$

1: Rewards group $\mathcal{R} = \{\}$
2: **for** $i = 0, 1, 2, \dots$ **do**
3:   Sample first pruned layer ID $l \sim \pi_{\theta_i}$ and construct ELM $\hat{\mathcal{F}}_l$
4:   Compute loss $\mathcal{L}_{\text{sft}}$ and rewards $\mathcal{L}_{\text{ce}}$ (see Eq. 9)
5:   $\mathcal{R}.\text{append}(-\mathcal{L}_{\text{ce}})$        ▷ Store negative cross-entropy as rewards
6:   Backpropagation($\mathcal{L}_{\text{sft}}$)
7:   Optimize($\hat{\mathcal{F}}_l$)
8:   **if** $i \mod p == 0$ **then**          ▷ Policy update phase
9:    **for** $j = 0, 1, \dots, q-1$ **do**
10:     Compute policy loss:

$$\mathcal{L}_{\text{policy}} \leftarrow \mathcal{L}_{\pi_{\theta_{\text{old}}}}^{\text{CLIP}}(\pi_\theta) \quad \text{(see Eq. 11)}$$

11:     Backpropagation($\mathcal{L}_{\text{policy}}$)
12:     Optimize($\theta$)
13:    **end for**
14:    $\mathcal{R}.\text{clear}()$           ▷ Reset rewards list for next cycle
15:   **end if**
16: **end for**
17: **return** $\theta$

---

where $p$ denotes the size of the reward set. The policy update follows

$$\mathcal{L}_{\pi_{\theta_{\text{old}}}}^{\text{CLIP}}(\pi_\theta) = \mathbb{E}_{l \sim \pi_\theta}[\min(\frac{\pi_\theta}{\pi_{\theta_{\text{old}}}}\hat{A}, \text{clip}(\frac{\pi_\theta}{\pi_{\theta_{\text{old}}}}, 1-\epsilon, 1+\epsilon)\hat{A})]. \tag{11}$$

with the advantage function defined as the normalized reward,

$$\hat{A} = \frac{\mathcal{R} - \text{mean}(\mathcal{R})}{\text{std}(\mathcal{R})}. \tag{12}$$

Training proceeds in cycles: rewards are collected over $p$ steps, and then the policy parameters are updated for $q$ optimization steps using the aggregated rewards. After convergence, the first pruned layer index is $\arg\max\theta$. Finally, pruning layers start from $\arg\max\theta$, and the equilibrium model is constructed according to the workflow described in Sec. 3.1. The paradigm of GPPO is illustrated in Fig. 3 and the pseudo-implementation is shown in Algo. 1.

### 3.3 ONE-STEP KV CACHE

With respect to the deployment of large language models at inference time, the memory overhead incurred by the KV cache is a significant challenge. With the conventional LLM inference pattern, the fixed-point module is executed many times iteratively to achieve convergence. In the $k$-th step, all previous KV caches are included in the calculation as

$$z_{T+1}^{(k+1)} = \varphi(K_{1\cdots T}^{(k)}, V_{1\cdots T}^{(k)}, z_{T+1}^{(k)}, h_{T+1}). \tag{13}$$

Consequently, the storage complexity of the Multi-Step KV cache in the fixed-point module is $\mathcal{O}(T \cdot$

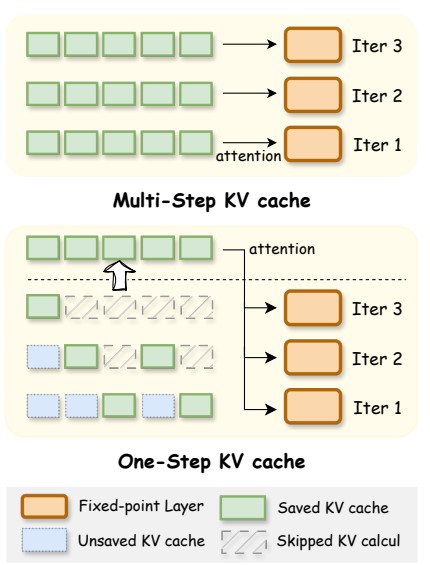

Figure 4: Illustration of cache strategies in the fixed-point layer. One-Step KV cache requires only the cache from the last iteration, while Multi-Step requires all caches.

$M$), where $T$ denotes the length of previous tokens and $M$ is the maximum number of fixed-point iterations. The One-Step KV cache stores and reuses only the cache from the last round in fixed-point iterations of all preceding tokens, which is then applied during the fixed-point iterations of the current token. Since intermediate features $z^{(k)}$ converge to the equilibrium state $z^*$, One-Step KV does not compromise the accuracy of the equilibrium language model and consistently converges to the approximate results as the Multi-Step KV cache under the appropriate conditions.

In the $k$-th fixed point step of the $T+1$ token, the One-Step KV cache iteration is formulated as

$$z_{T+1}^{(k+1)} = \varphi(\boldsymbol{K}_{1\cdots T}^*, \boldsymbol{V}_{1\cdots T}^*, z_{T+1}^{(k)}, \boldsymbol{h}_{T+1}). \tag{14}$$

Under this framework, the storage complexity of the KV cache becomes $\mathcal{O}(T)$. Moreover, it eliminates issues related to KV cache missing when allocating different iteration steps among tokens, enabling the efficient deployment of fixed-point solvers to accelerate our proposed ELMs.

## 3.4 FIXED-POINT SOLVING ACCELERATION

Benefiting from the One-Step KV cache mechanism, our method can efficiently adopt an adaptive termination criterion for the fixed-point iterations: the process is halted once the $\ell_2$ norm of the residual between successive states, $\|\varphi(z^{(k)}, \boldsymbol{h}) - z^{(k)}\|_2$, falls below a predefined threshold $\delta$. The final output is then taken as the equilibrium state $z^*$. This approach allows variable computational budgets across tokens, depending on their individual convergence speed.

In addition to the standard fixed-point (simple) iteration, we consider two classical acceleration techniques to further reduce the required iterations. Broyden's method (Broyden, 1965) employs a quasi-Newton update that iteratively approximates the inverse Jacobian. Anderson acceleration (Walker & Ni, 2011) uses a linear combination of previous iterations to extrapolate.

## 4 EXPERIMENTS

### 4.1 EXPERIMENTAL SETTINGS

**Source Model.** We evaluate our method on popular open-source LLMs: Qwen2.5-1.5B-Instruct, Qwen2.5-7B-Instruct (Qwen et al., 2025) and Llama3.2-3B-Instruct (Dubey et al., 2024). All models consist of 28 transformer layers. We use their finetuned counterparts as baselines and compare against them to measure the performance drop caused by parameter pruning. Additional implementation details are provided in A.7.

**Baselines.** We compare against representative pruning baselines. Sheared-LLaMA (Xia et al., 2023) applies structured pruning over heads, FFN channels, hidden dimensions, and layers with $l_0$ regularization and LoRA finetuning. Shortened-LLaMA (Kim et al., 2024) prunes layers by perplexity. ReplaceMe (Shopkhoev et al., 2025) and LLM-Streamline (Chen et al., 2024) replace consecutive layers with lightweight modules based on cosine similarity. ReplaceMe utilizes linear transformation for feature alignment and LLM-Streamline utilizes a transformer layer or an FFN module.

**Benchmarks.** We conduct on three downstream tasks, including: commonsense reasoning, mathematical problem solving, and code generation. We fine-tune the model on PIQA (Bisk et al., 2020), SIQA (Sap et al., 2019), OBQA (Mihaylov et al., 2018) and MMLU (Hendrycks et al., 2020) to verify commonsense reasoning; MetaMathQA (Yu et al., 2023) for mathematical problem solving; and OpenCoder (Huang et al., 2024) for code generation. Evaluation results are performed in zero-shot settings among PIQA, SIQA, OBQA, MMLU, GSM8K (Cobbe et al., 2021), MATH (Hendrycks et al., 2021), HumanEval (Chen et al., 2021), and MBPP (Austin et al., 2021).

### 4.2 QUANTITATIVE ANALYSIS

**Comparisons with State-of-the-Arts.** We present the performance of all benchmarks compared with sota method under the same settings. With pruning eight layers, *i.e.*, $M{=}8$ (about 28.6% of the non-embedding parameters compressed), ELMs consistently outperform baseline methods in both compression efficiency and stability as shown in Table 1. On average, they preserve 99% the precision of fine-tuned dense LLMs across benchmarks and base models.

Table 1: Performance comparison of pruning baseline methods on commonsense, math, and code benchmarks. The last columns reports the average accuracy across all benchmarks and the relative performance (RP) compared with dense models.

| Model | Method | Ratio | Commonsense | | | | Math | | Code | | Average | RP |
|---|---|---|---|---|---|---|---|---|---|---|---|---|
| | | | PIQA | SIQA | OBQA | MMLU | GSM8K | MATH | HumanEval | MBPP | | |
| Qwen2.5-1.5B-Instruct | Dense | 0.00% | 83.4 | 77.1 | 85.4 | 54.8 | 72.4 | 32.2 | 73.8 | 71.8 | 68.9 | 100.0% |
| | Sheared-LLaMA | 28.5% | 70.6 | 67.0 | 68.0 | 37.6 | 69.6 | 29.6 | 75.0 | 69.0 | 60.8 | 88.3% |
| | Shortened-LLaMA | 28.6% | 75.6 | 71.8 | 74.8 | 41.5 | 63.0 | 24.5 | 64.0 | 65.6 | 60.1 | 87.3% |
| | ReplaceMe(Cosine) | 28.6% | 57.3 | 41.2 | 34.4 | 28.8 | 0.00 | 0.00 | 1.83 | 2.00 | 20.7 | 30.0% |
| | ReplaceMe(LS) | 28.6% | 57.9 | 40.9 | 33.8 | 29.1 | 0.00 | 0.00 | 1.22 | 1.20 | 20.5 | 29.8% |
| | LLM-Streamline(Layer) | 28.6% | 75.6 | 70.7 | 71.6 | 45.2 | 44.3 | 16.1 | 38.4 | 53.4 | 51.9 | 75.4% |
| | LLM-Streamline(FFN) | 28.6% | 70.6 | 58.8 | 58.0 | 37.3 | 24.4 | 10.4 | 28.7 | 44.6 | 41.6 | 60.4% |
| | ELM(Ours) | 28.6% | 83.1 | 76.9 | 83.8 | 54.0 | 71.8 | 30.5 | 74.4 | 70.6 | 68.1 | 98.9% |
| Llama3.2-3B-Instruct | Dense | 0.00% | 83.8 | 77.9 | 79.6 | 51.0 | 75.4 | 33.0 | 78.0 | 70.0 | 68.6 | 100.0% |
| | Sheared-LLaMA | 28.3% | 77.1 | 73.6 | 68.2 | 39.1 | 68.4 | 27.2 | 70.7 | 69.8 | 61.8 | 90.0% |
| | Shortened-LLaMA | 28.6% | 82.6 | 77.1 | 79.2 | 49.7 | 67.6 | 25.4 | 73.2 | 65.4 | 65.4 | 95.4% |
| | ReplaceMe(Cosine) | 28.6% | 82.5 | 77.6 | 73.0 | 45.3 | 0.08 | 0.34 | 0.00 | 0.00 | 34.9 | 50.8% |
| | ReplaceMe(LS) | 28.6% | 82.7 | 77.8 | 72.6 | 45.0 | 0.30 | 0.36 | 0.00 | 0.00 | 34.8 | 50.8% |
| | LLM-Streamline(Layer) | 28.6% | 83.9 | 77.9 | 79.6 | 50.8 | 70.4 | 26.3 | 76.2 | 67.0 | 66.5 | 97.0% |
| | LLM-Streamline(FFN) | 28.6% | 84.1 | 78.0 | 79.8 | 50.6 | 63.6 | 22.5 | 66.5 | 63.8 | 63.6 | 92.7% |
| | ELM(Ours) | 28.6% | 84.0 | 78.4 | 79.6 | 50.5 | 74.5 | 29.7 | 76.8 | 69.8 | 67.9 | 99.0% |
| Qwen2.5-7B-Instruct | Dense | 0.00% | 87.4 | 79.2 | 87.0 | 60.2 | 82.6 | 42.4 | 88.4 | 78.4 | 75.7 | 100.0% |
| | Sheared-LLaMA | 28.5% | 81.8 | 77.0 | 78.4 | 47.3 | 68.0 | 25.4 | 81.7 | 72.6 | 66.5 | 87.9% |
| | Shortened-LLaMA | 28.6% | 86.3 | 79.1 | 84.4 | 58.4 | 65.4 | 25.6 | 71.3 | 68.2 | 67.3 | 89.0% |
| | ReplaceMe(Cosine) | 28.6% | 57.4 | 49.9 | 41.6 | 32.1 | 0.00 | 0.00 | 11.0 | 11.8 | 25.5 | 33.6% |
| | ReplaceMe(LS) | 28.6% | 57.4 | 49.9 | 41.6 | 32.1 | 0.00 | 0.00 | 11.0 | 11.8 | 25.5 | 33.6% |
| | LLM-Streamline(Layer) | 28.6% | 78.0 | 73.1 | 78.2 | 45.1 | 53.4 | 22.6 | 53.0 | 59.6 | 57.9 | 76.5% |
| | LLM-Streamline(FFN) | 28.6% | 72.7 | 68.5 | 72.0 | 42.4 | 35.4 | 15.8 | 40.2 | 53.4 | 50.1 | 66.1% |
| | ELM(Ours) | 28.6% | 86.3 | 79.3 | 87.0 | 59.2 | 82.2 | 41.1 | 88.4 | 76.2 | 75.0 | 99.0% |

Unlike baselines that collapse on mathematical reasoning and code generation tasks, ELMs retain at least 90% accuracy even on MATH. For example, on GSM8K, the ELM on Qwen2.5-7B-Instruct achieves 99.5% accuracy of dense, outperforms Sheared-LLaMA by 17.2%, Shortened-LLaMA by 20.4% and LLM-Streamline by 34.9%, while ReplaceMe fails completely. Performance gains also hold across models. Whereas methods like LLM-Streamline maintain high accuracy only on Llama3.2-3B-Instruct, their performance drops sharply on Qwen2.5. In contrast, ELMs deliver consistent improvements, e.g., surpassing LLM-Streamline by 23.5% on Qwen2.5-1.5B-Instruct and 22.5% on Qwen2.5-7B-Instruct in average. These results highlight the robustness and general applicability of our method.

**Performance under Different Compression Ratios.** We evaluate ELMs with varying numbers of pruned layers $M=8, 10, 12, 14$ on Qwen2.5-1.5B-Instruct and Llama3.2-3B-Instruct for code synthesis and commonsense reasoning. As shown in Table 2, ELM consistently outperforms pruning baselines across models and benchmarks under the same pruning ratio. With increasing pruning ratio, baselines suffer severe degradation, while ELM exhibits much milder decline, highlighting its robustness. On Llama3.2-3B-Instruct, ELM still retains 97.7% of the dense performance on MBPP with 50% layers pruned, outperforming Shortened-LLaMA by 8.6% and LLM-Streamline by 43.4%. For MMLU, ELM retains 98.6% accuracy, outperforming Sheared-LLaMA by 35.7%.

Table 2: Comparison of different numbers of pruned layers for code and commonsense tasks.

| Model | Method | HumanEval | | | | MBPP | | | | MMLU | | | |
|---|---|---|---|---|---|---|---|---|---|---|---|---|---|
| | | $M$ | | | | $M$ | | | | $M$ | | | |
| | | 8 | 10 | 12 | 14 | 8 | 10 | 12 | 14 | 8 | 10 | 12 | 14 |
| Qwen2.5-1.5B-Instruct | Sheared-LLaMA | 75.0 | 68.3 | 62.2 | 50.0 | 69.0 | 67.6 | 65.2 | 60.4 | 37.6 | 40.5 | 38.2 | 32.1 |
| | Shortened-LLaMA | 64.0 | 59.1 | 53.0 | 46.3 | 65.6 | 61.8 | 60.2 | 58.2 | 41.5 | 39.4 | 36.9 | 34.6 |
| | LLM-Streamline(Layer) | 38.4 | 32.3 | 22.0 | 18.9 | 53.4 | 47.6 | 41.2 | 37.2 | 45.2 | 39.0 | 36.1 | 34.1 |
| | ELM(Ours) | 74.4 | 71.3 | 66.5 | 58.5 | 70.6 | 69.4 | 67.4 | 65.4 | 54.0 | 53.0 | 50.2 | 46.2 |
| Llama3.2-3B-Instruct | Sheared-LLaMA | 69.0 | 67.5 | 66.9 | 65.5 | 69.8 | 67.3 | 66.8 | 67.1 | 39.1 | 40.3 | 39.3 | 38.1 |
| | Shortened-LLaMA | 73.2 | 67.7 | 64.6 | 61.0 | 68.6 | 69.6 | 66.0 | 62.4 | 49.7 | 50.1 | 49.5 | 46.2 |
| | LLM-Streamline(Layer) | 76.2 | 75.0 | 23.8 | 22.6 | 67.0 | 64.0 | 42.0 | 38.0 | 50.8 | 50.8 | 50.1 | 32.1 |
| | ELM(Ours) | 76.8 | 77.4 | 73.2 | 68.9 | 69.8 | 71.0 | 70.4 | 68.4 | 50.5 | 50.8 | 50.7 | 50.3 |

### 4.3 STUDIES ON GROUP PRUNING POLICY OPTIMIZATION

In this section, we compare our proposed Group Pruning Policy Optimization (GPPO) in Sec. 3.2 with statistical metrics of **cosine similarity** and **perplexity**. We randomly sample $3k$ instances from the training set to calibrate the cosine similarity and perplexity metrics. GPPO is trained for two epochs on each task. To illustrate the impact of pruning strategies, we report the results of a commonsense task (MMLU) and a code generation task (MBPP) under different choices of $l^*$ for ELM as shown in Fig. 5. The results show that pruning strategies are critical and the optimal $l^*$ varies among downstream tasks. GPPO consistently identifies the best choice of $l^*$, matching the outcomes of enumeration experiments. Moreover, the learned values $\theta$ exhibit a stronger positive correlation with the performance of ELMs, confirming the effectiveness of our approach.

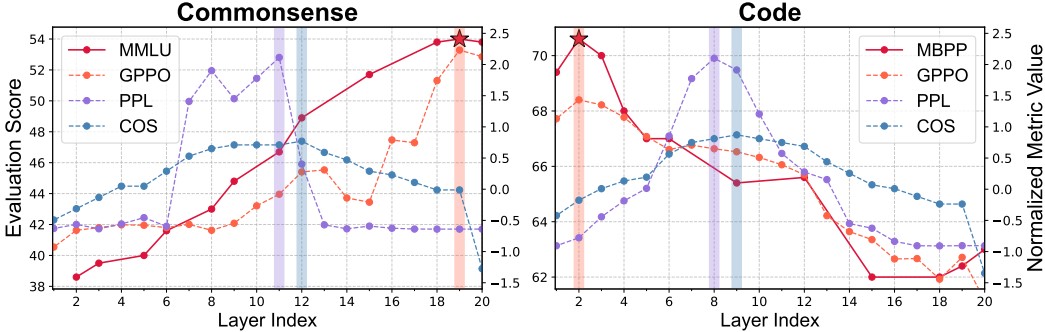

Figure 5: Layer-wise experimental values on ELMs and normalized pruning measures for common and code tasks. A colored rectangle stands for the maximal value of each metric, *i.e.*, cosine similarity, perplexity and GPPO. The best pruning performance of ELM is signed as a red star.

### 4.4 STUDIES ON ONE-STEP KV CACHE

As demonstrated in Sec. 3.3, we show the effectiveness of the One-Step KV in fixed-point solving converge to the same results as those obtained with the Multi-Step KV cache. We conduct ablation studies on ELMs with and without the One-Step KV cache under different pruning settings ($M{=}8, 14$). We employ simple iteration as the solver and set the number of iterations equal to $M$. Table 3 reports both the accuracy of ELMs on Qwen2.5-1.5B-Instruct across mathematical reasoning benchmarks and KV cache consumption at a 4K context length in 16-bit. The results show that the One-Step KV cache achieves nearly identical performance to the Multi-Step variant, confirming the theoretical equivalence. Moreover, it reduces memory overhead by 25% and 46% for $M{=}8$ and $M{=}14$, respectively, thereby substantially lowering the memory footprint during inference.

Table 3: Memory efficiency of ELM under different $M$.

| $M$ | Method | Iters | Model(GB) | KV(MB) | GSM8K (Acc.) | MATH (Acc.) |
|---|---|---|---|---|---|---|
| – | Dense | – | 3.09 | 114.8 | 72.4 | 32.2 |
| 8 | ELM (w/o One-Step KV) | 8 | 2.44 | 114.8 | 71.4 | 30.8 |
|   | ELM (w/  One-Step KV) | 8 | 2.44 | 86.0 | 71.8 | 30.5 |
| 14 | ELM (w/o One-Step KV) | 14 | 1.88 | 114.8 | 63.3 | 24.1 |
|    | ELM (w/  One-Step KV) | 14 | 1.88 | 61.6 | 63.6 | 24.0 |

### 4.5 STUDIES ON ACCELERATION SOLVERS

Leveraging the fixed-point property of ELMs with the One-Step KV cache, we apply existing numerical solvers to accelerate convergence in fixed-point iterations and reduce computational cost. We evaluate this approach on GSM8K with Qwen2.5-1.5B-Instruct under pruning settings ($M{=}8, 14$),

considering simple iteration, Broyden's method, and Anderson acceleration. Table 4 shows the performance and iteration counts under different $\delta$ values.

Overall, the accuracy exhibits minimal variation across different $\delta$ values. Among different solvers, Broyden's method shows weaker performance when $M=8$, due to larger errors in approximating the inverse Jacobian. Anderson acceleration achieves the most stable speedup across pruning depths, e.g., reducing computation cost by 38.6% over simple iteration when $M$ is set to 14, and $\delta$ is 0.5 with negligible accuracy loss. Even with a loose $\delta$ ($\delta=4.0$), ELMs retains over 98% and 86% accuracy of the dense LLMs with 8 and 14 pruned layers, respectively. At the same time, ELMs reduce computation cost by 17% and 35%. These results demonstrate that ELMs not only shrink parameter size but also substantially reduce computational overhead, enabling faster inference.

Table 4: Performance comparison of solvers under different thresholds $\delta$ on GSM8K benchmark with Qwen2.5-1.5B-Instruct. Rows indicate methods (simple, broyden, anderson), while columns correspond to thresholds.

|  | Method | $M = 8$ | | | | | | $M = 14$ | | | | | |
|  |  | Threshold $\delta$ | | | | | | Threshold $\delta$ | | | | | |
|  |  | 0.00 | 0.50 | 2.00 | 4.00 | 10.0 | 100.0 | 0.00 | 0.50 | 2.00 | 4.00 | 10.0 | 100.0 |
| Acc. | simple | 71.8 | 71.3 | 72.0 | 71.6 | 70.1 | 68.8 | 63.3 | 63.8 | 63.2 | 63.8 | 62.5 | 35.8 |
|  | broyden | 69.8 | 70.5 | 69.7 | 70.1 | 70.1 | 68.9 | 63.2 | 62.9 | 63.5 | 61.0 | 59.0 | 36.2 |
|  | anderson | 71.6 | 71.0 | 71.0 | 71.1 | 70.4 | 68.8 | 63.4 | 64.3 | 62.7 | 62.6 | 60.3 | 35.5 |
| Iters | simple | 8.00 | 7.16 | 4.60 | 3.60 | 2.43 | 1.06 | 14.00 | 12.25 | 7.03 | 5.52 | 3.63 | 1.10 |
|  | broyden | 8.00 | 7.22 | 5.10 | 4.16 | 2.43 | 1.06 | 14.00 | 8.05 | 5.47 | 4.55 | 3.49 | 1.10 |
|  | anderson | 8.00 | 5.56 | 4.08 | 3.37 | 2.43 | 1.06 | 14.00 | 7.52 | 5.18 | 4.27 | 3.16 | 1.10 |

## 4.6 Inference Efficiency

**Memory Consumption.** We further evaluate the inference-time memory efficiency of ELM under realistic deployment settings. The primary memory cost of LLM inference stems from model parameters and the KV cache. Table. 3 reports the memory consumption of ELM based on Qwen2.5-1.5B-Instruct. Compared with Dense baseline, ELMs with $M=8$ and $M=14$ pruned layers reduce the parameter memory from 3.09 GB to 2.44 GB and 1.88 GB, achieving 21% and 39% savings, respectively. When combined with our One-Step KV cache, ELM further decreases KV cache overheads at a 4K context length from 114.7 MB to 86.0 MB (25% reduction) for $M=8$ and to 61.6 MB (46% reduction) for $M=14$. These results confirm that ELM substantially lowers the total memory footprint and can benefit deployment on memory-constrained platforms.

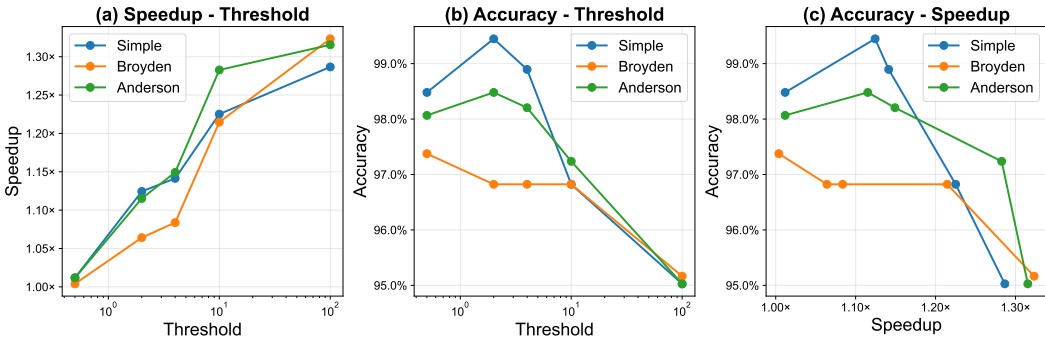

Figure 6: Performance of solvers on ELMs based on Qwen2.5-1.5B-Instruct across various thresholds. The decode speed and accuracy of GSM8K is normalized with Dense.

**Decoding Speedup.** We also evaluate the computational efficiency of ELM during decoding. Although ELM relies on fixed-point iterations, it achieves substantial decoding speedup due to two

key reasons: (1) appropriate choices of the convergence threshold $\delta$ and acceleration methods effectively reduce iteration counts, and (2) the computational cost of the solvers and convergence checks is negligible. As shown in Fig. 6, ELM with $M{=}8$ accelerates compared with the dense model across different solvers with thresholds $\delta{\geq}0.5$. With $\delta{=}4$, the Anderson solver converges in only 2.4 iterations and decodes at 87.8 token/s, while the dense model only reaches 76.4 token/s. Fig. 6 (c) illustrates the accuracy-speedup Pareto frontier, where $\delta{=}10$ of Anderson solver provides a favorable balance, achieving 28.3% speedup while maintaining 97% accuracy. Even under an aggressive setting of $\delta{=}100$, ELM continues to deliver usable performance. The effect is attributed to SJFB training, which enables stable generation even with few iteration steps. These results demonstrate that fixed-point iteration is not a decoding bottleneck: iteration counts remain small, solver overhead is minimal, and the accuracy-speedup trade-off can be smoothly tuned via the threshold $\delta$.

## 5 RELATED WORK

**Layer-Level Pruning in LLMs Research.** Previous pruning approaches for LLMs have mainly focused on weight matrices (Frantar & Alistarh, 2023; Sun et al., 2023), attention heads, and hidden dimensions (Ma et al., 2023; Xia et al., 2023), but these often introduce structural irregularities. More recently, layer-level pruning has emerged as a promising alternative due to its deployment efficiency across platforms. For example, Shortened-LLaMA (Kim et al., 2024) removes layers based on metrics such as perplexity or Taylor expansion and employs LoRA fine-tuning for recovery. LLM-Streamline (Chen et al., 2024) and ReplaceMe (Shopkhoev et al., 2025) prune consecutive intermediate layers and insert lightweight alignment modules: the former reintroduces a transformer or feed-forward block, whereas the latter applies a simple linear mapping. Our method follows this paradigm but employs a novel alignment network, achieving substantially higher performance.

**Looped Mechanisms in LLMs Research.** Standard LLMs consist of stacked residual transformer layers, while recent studies (Zhu et al., 2025; Chen et al., 2025; Bae et al., 2025; Li et al., 2025; Geiping et al., 2025) investigate looped architectures. For instance, Inner Thinking Transformer (Chen et al., 2025) routes every even-numbered layer through a learned loop, and Mixture-of-Recursion (Bae et al., 2025) generalizes this with diverse recursive designs. ELMs advance this line of work with two key innovations: (1) grounding looped computation in fixed-point theory, which enables train-free acceleration solvers and adaptive per-token iteration; and (2) pioneering the use of looped mechanisms for post-training LLM compression.

## 6 CONCLUSION

We introduced Equilibrium Language Models (ELMs), a novel LLM compression framework that replaces consecutive transformer layers with a fixed-point network and leverages a policy optimization framework GPPO to learn optimal pruning strategies. By exploiting One-Step KV cache and acceleration solvers, ELMs effectively reduce computational overheads while maintaining model fidelity. Experiments demonstrate that ELMs retain over 99% of the performance of dense models on average with 28% fewer parameters, and outperform state-of-the-art pruning methods on complex generation tasks such as mathematical reasoning and code synthesis.

In addition, we discuss several directions for future work. Our experiments primarily focus on post-training on downstream tasks, leaving the generalization of ELMs in pre-training scenarios open for investigation. Furthermore, our current design only prunes the single consecutive layer interval. Extending ELMs to more aggressive architectures such as transformers composed entirely of fixed-point layers offers a promising avenue for future research.

## REPRODUCIBILITY STATEMENT

We have made extensive efforts to ensure the reproducibility of our results. Details of the proposed model and algorithms are provided in Sec. 3, with additional related algorithms, derivations and proofs given in A. All base LLMs and datasets used in our experiments are publicly available. The experimental setup, including hyperparameter choices, data processing, evaluation protocols, and baseline implementation details are described in Sec. 4, and further elaborated in Appendix A.7. We provide the core implementation for reproducing our method in https://github.com/Jyk-122/ELM.

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

# A APPENDIX

## A.1 STOCHASTIC JACOBIAN-FREE BACKPROPAGATION

We present the pseudo code of SJFB method in Sec. 3.1 as follows.

---
**Algorithm 2** Stochastic Jacobian-Free Backpropagation
---
**Require:** input hidden states $\boldsymbol{h}^l$, target hidden states $\boldsymbol{h}^{l+M}$,
**Hyperparameters:** Max iterations $M$

1: **function** TRAINSTEP($\hat{\mathcal{F}}_l$)
2:     $\boldsymbol{z} \leftarrow$ zero_like($\boldsymbol{h}^l$)
3:     **for** $m$ sampled uniformly from 0 to $M/2$ **do**        ▷ Not included in backpropagation
4:         $\boldsymbol{z} \leftarrow \varphi(\boldsymbol{z}, \boldsymbol{h}^l)$
5:     **end for**
6:     $\boldsymbol{z} \leftarrow \boldsymbol{z}$.detach()
7:     **for** $n$ sampled uniformly from 1 to $M/2$ **do**
8:         $\boldsymbol{z} \leftarrow \varphi(\boldsymbol{z}, \boldsymbol{h}^l)$
9:     **end for**
10:     Compute loss $\mathcal{L}_{\text{sft}}(\hat{\mathcal{F}}_l, \boldsymbol{h}^{l+m}, \boldsymbol{z})$ (see Eq. 9)
11:     Backpropagation($\mathcal{L}_{\text{sft}}$)
12:     Optimize($\hat{\mathcal{F}}_l$)
13: **end function**
---

## A.2 GPPO

**Baseline Metrics.** We formulate the baseline metrics(cosine similarity and perplexity) in Sec.4.3:

$$l^*_{\text{COS}} = \arg\max_l \mathbb{E}_{\boldsymbol{h}\sim\mathcal{C}} \frac{\|\boldsymbol{h}^l \cdot \boldsymbol{h}^{l+M}\|}{\|\boldsymbol{h}^l\|\|\boldsymbol{h}^{l+M}\|}, \tag{15}$$

$$l^*_{\text{PPL}} = \arg\max_l \mathbb{E}_{x\sim\mathcal{C}} \exp\{-\frac{1}{T}\sum_t \log p_{\tilde{\mathcal{F}}_{[l,l+M-1]}}(x_t|x_{<t})\}, \tag{16}$$

where $\boldsymbol{h}$ is the hidden states, $x$ is input tokens and $\mathcal{C}$ is the calibration dataset. $\tilde{\mathcal{F}}_{[l,l+M-1]}$ represents LLM $\mathcal{F}$ with $M$ consecutive layers pruned starting from $l$ and $T$ is the length of context.

**Stability.** We visualize the loss curves of GPPO training based on Qwen2.5-1.5B-Instruct under code synthesis task. Fig. 7 shows that GPPO successfully learns the best layer pruning index ($l^* = 2$) with increasing rewards.

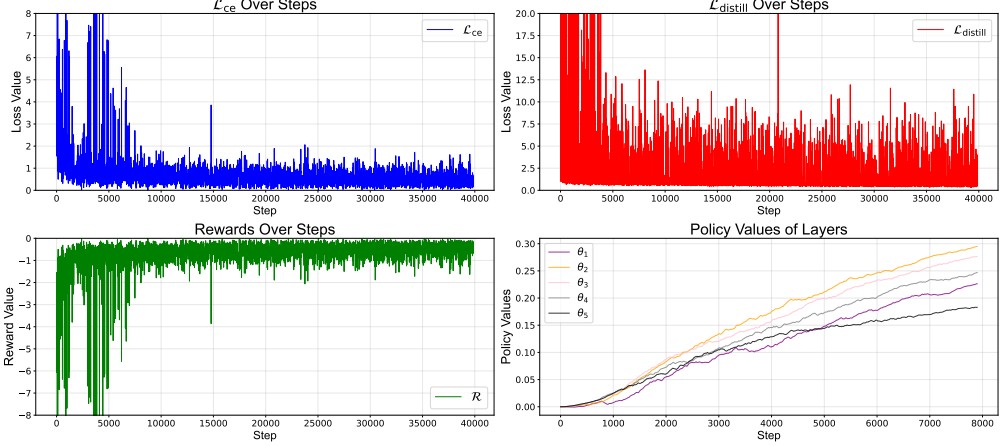

Figure 7: GPPO loss curves, including the cross entropy, distillation, rewards and the value of policy parameters.

**Costs.** We analyze the overhead introduced by GPPO in Table 5. Given the calibration dataset with size $10k$, training ELM with $M = 8$ pruned layers on Qwen2.5-1.5B-Instruct for 4 epochs requires 0.15 GPU·days with 17.5 GB memory. Enumerating all possible start layers would therefore cost 3.15 GPU·days, which is clearly expensive. GPPO circumvents this issue through a lightweight policy network with LoRA-based updates: with only 2 epochs of training, it converges to the optimal start layer in 0.07 GPU·days, using 21.5 GB memory. Compared with heuristic metrics such as Perplexity and Cosine Similarity, computational costs are similar but resulted configurations lead to substantially worse accuracy while GPPO provides an efficient and reliable solution.

Table 5: Training cost and pruning performance of GPPO compared with other metrics.

| Method | Time (GPU·days) | Memory (GB) | Start Layer | MBPP |
|---|---|---|---|---|
| ELM training (single) | 0.15 | 17.5 | – | – |
| Enumeration | 3.15 | 17.5 | 2 | 70.6 |
| GPPO | 0.07 | 21.5 | 2 | 70.6 |
| PPL | 0.99 | 10.1 | 8 | 66.6 |
| Cos. | 0.03 | 8.7 | 9 | 65.4 |

## A.3 ABLATIONS ON ELM

**Ablation on Fixed-Point Module.** We conduct ablation study on fixed-point layer module in commonsense reasoning task. ELM(FFN) behaves significant performance degeneration compared with ELM(Layer), indicating the necessity of using a transformer layer to compress the pruned module.

Table 6: Commonsense reasoning performance on different fixed-point module designs.

| | PIQA | SIQA | OBQA | MMLU |
|---|---|---|---|---|
| ELM(FFN) | 82.6 | 74.7 | 81.2 | 51.6 |
| ELM(Layer) | **83.1** | **76.9** | **83.8** | **54.0** |

**Ablation on $W_h$ initialization.** We present the distillation loss value in the first $4k$ training steps on coding task. Results shows that initializing $W_h$ as identity significantly accelerates the convergence of ELM training compared with random and also achieves better performance in MBPP.

Table 7: Distillation loss and MBPP performance of different initializations of $W_h$.

| $W_h$ | 0 | $1k$ | $2k$ | $3k$ | $4k$ | MBPP Acc. |
|---|---|---|---|---|---|---|
| random | 210.000 | 0.824 | 0.459 | 0.398 | 0.355 | 69.8 |
| identity | 2.172 | 0.272 | 0.198 | 0.152 | 0.154 | **70.6** |

**Ablation on SJFB**. We compare the training of ELM with or without SJFB based on Qwen2.5-1.5B-Instruct in math task. The dataset size is $395k$ with 4 epochs. ELM without SJFB costs 7.1 GPU·days with 20.1 GB memory and ELM with SJFB costs 6.1 GPU·days with 17.5 GB, reducing about 14% training time and 13% memory cost.

**Base LLMs with different depths.** We provide experiments over commonsense reasoning task and code synthesis based on Qwen2.5-3B-Instruct, which contains 36 layers and we prune 10 layers. Table 8 shows that ELM still achieves 98.7% performance of Dense in average.

Table 8: Performance of ELM based on Qwen2.5-3B-Instruct with $M = 10$.

| | Ratio | PIQA | SIQA | OBQA | MMLU | GSM8K | MATH | HumanEval | MBPP | Avg. | RP |
|---|---|---|---|---|---|---|---|---|---|---|---|
| Dense | 0.00% | 85.7 | 80.7 | 87.0 | 61.0 | 77.4 | 37.5 | 87.8 | 75.8 | 74.1 | 100.0% |
| ELM | 27.8% | 85.3 | 80.5 | 84.8 | 60.5 | 78.7 | 36.6 | 86.0 | 73.0 | 73.2 | 98.7% |

## A.4 CONSISTENCY OF ONE-STEP KV CACHE

Under appropriate circumstances in ELMs, inference with One-Step KV cache (Eq. 14) converges to the same result as Multi-Step KV cache (Eq. 13). We introduce the proposition of consistency between two methods as follows.

**Proposition 1.** If the fixed-point function $\varphi$ satisfies that the norm of derivatives corresponding to key and value tensors $\frac{\partial \varphi}{\partial K}, \frac{\partial \varphi}{\partial V}$ is bounded, fixed-point iteration with One-Step KV cache (Eq. 14) converges to the same result as iteration with Multi-Step KV cache (Eq. 13).

*Proof.* We denote $z_{T+1}^*$ as the convergent value with Multi-Step KV cache

$$z_{T+1}^* = \lim_{k \to \infty} \varphi(K_{1\dots T}^{(k)}, V_{1\dots T}^{(k)}, z_{T+1}^{(k)}, h_{T+1}), \tag{17}$$

Let $K_{1\dots T}^*, V_{1\dots T}^*$ represent the convergent value of KV cache:

$$\lim_{k \to \infty} K_{1\dots T}^{(k)} = K_{1\dots T}^* \\ \lim_{k \to \infty} V_{1\dots T}^{(k)} = V_{1\dots T}^* \tag{18}$$

The sequence of results with One-Step KV cache is formulated as

$$z_{T+1}^{(k+1)} = \varphi(K_{1\dots T}^*, V_{1\dots T}^*, z_{T+1}^{(k)}, h_{T+1}). \tag{19}$$

Considering the $\ell_2$-norm difference between two iteration sequences:

$$\|z_{T+1}^{(k+1)} - z_{T+1}^*\| = \|\varphi(K_{1\dots T}^*, V_{1\dots T}^*, z_{T+1}^{(k)}, h_{T+1}) - z_{T+1}^*\| \tag{20}$$

According to the Triangle Inequality, we have

$$\|z_{T+1}^{(k+1)} - z_{T+1}^*\| \leq \|\varphi(K_{1\dots T}^*, V_{1\dots T}^*, z_{T+1}^{(k)}, h_{T+1}) - \varphi(K_{1\dots T}^{(k)}, V_{1\dots T}^{(k)}, z_{T+1}^{(k)}, h_{T+1})\| \\ + \|\varphi(K_{1\dots T}^{(k)}, V_{1\dots T}^{(k)}, z_{T+1}^{(k)}, h_{T+1}) - z_{T+1}^*\| \tag{21}$$

For the first term, we use first-order Taylor expansion

$$\|\varphi(K_{1\dots T}^*, V_{1\dots T}^*, z_{T+1}^{(k)}, h_{T+1}) - \varphi(K_{1\dots T}^{(k)}, V_{1\dots T}^{(k)}, z_{T+1}^{(k)}, h_{T+1})\| \\ = \|\frac{\partial \varphi}{\partial K}(K_{1\dots T}^* - K_{1\dots T}^{(k)})\| + \|\frac{\partial \varphi}{\partial V}(V_{1\dots T}^* - V_{1\dots T}^{(k)})\| + R^{(k)} \\ \leq \|\frac{\partial \varphi}{\partial K}\|\|(K_{1\dots T}^* - K_{1\dots T}^{(k)})\| + \|\frac{\partial \varphi}{\partial V}\|\|(V_{1\dots T}^* - V_{1\dots T}^{(k)})\| + R^{(k)} \tag{22}$$

where $R^{(k)} = o(\|K_{1\dots T}^* - K_{1\dots T}^{(k)}\|) + o(\|V_{1\dots T}^* - V_{1\dots T}^{(k)}\|)$ represents the remainder term. Given the condition that the norm of derivatives $\frac{\partial \varphi}{\partial K}, \frac{\partial \varphi}{\partial V}$ is bounded and the convergency of KV cache in Eq. 18, the limitations of the first term is zero:

$$\lim_{k \to \infty} \|\varphi(K_{1\dots T}^*, V_{1\dots T}^*, z_{T+1}^{(k)}, h_{T+1}) - \varphi(K_{1\dots T}^{(k)}, V_{1\dots T}^{(k)}, z_{T+1}^{(k)}, h_{T+1})\| = 0 \tag{23}$$

For the second term, due to the convergency in Eq. 17, we can give the limitations

$$\lim_{k \to \infty} \|\varphi(K_{1\dots T}^{(k)}, V_{1\dots T}^{(k)}, z_{T+1}^{(k)}, h_{T+1}) - z_{T+1}^*\| = 0. \tag{24}$$

Overall, the limitations of the difference is

$$\lim_{k \to \infty} \|z_{T+1}^{(k+1)} - z_{T+1}^*\| = 0, \tag{25}$$

which means that iteration with One-Step KV cache (Eq. 14) converges to the same result as iteration with Multi-Step KV cache (Eq. 13). $\square$

## A.5 ALGORITHMS OF SOLVERS

**Simple iteration.** The simple solver loops the fixed-point function until meets the equilibrium state.

---

**Algorithm 3** Simple iteration

---

**Require:** Function $\varphi(\cdot, \boldsymbol{h})$, init $\boldsymbol{z}^{(0)}$, tol $\delta$, max iters $M$
**Ensure:** Equilibrium $\boldsymbol{z}^*$
1: $k \leftarrow 0$
2: **while** $k < M$ **do**
3:    $\boldsymbol{z}^{(k+1)} \leftarrow \varphi(\boldsymbol{z}^{(k)}, \boldsymbol{h})$
4:    **if** $\|\boldsymbol{z}^{(k+1)} - \boldsymbol{z}^{(k)}\|_2 < \delta$ **then**
5:        **return** $\boldsymbol{z}^* = \boldsymbol{z}^{(k+1)}$
6:    **end if**
7:    $k \leftarrow k + 1$
8: **end while**
9: **return** $\boldsymbol{z}^* = \boldsymbol{z}^{(M)}$

---

**Broyden's method.** The Broyden solver is a quasi-Newton approach for solving nonlinear fixed-point equations, where the Jacobian (or its inverse) is iteratively approximated by low-rank updates. In our implementation, we approximate the inverse Jacobian matrix and omit the step-size search procedure, e.g., line search, as it typically involves an uncontrollable number of function evaluations. We adopt a fixed step size of 1 by default.

---

**Algorithm 4** Broyden's Method

---

**Require:** Function $g(\boldsymbol{z}) = \varphi(\boldsymbol{z}, \boldsymbol{h}) - \boldsymbol{z}$, init $\boldsymbol{z}^{(0)}$, max iters $M$, tol $\delta$
**Ensure:** Equilibrium $z^*$
1: Initialize $\boldsymbol{U}, V \leftarrow \emptyset$          ▷ low-rank factors of inverse Jacobian
2: $k \leftarrow 0$
3: $\boldsymbol{g}^{(0)} \leftarrow g(\boldsymbol{z}^{(0)})$
4: **while** $k < M - 1$ **do**
5:    **if** $\|\boldsymbol{g}^{(k)}\|_2 < \delta$ **then**
6:        **return** $\boldsymbol{z}^* = \boldsymbol{z}^{(k)}$
7:    **end if**
8:    Compute direction:
$$\boldsymbol{p}^{(k)} = -\big(-I + \boldsymbol{U}\boldsymbol{V}^\top\big)\boldsymbol{g}^{(k)}$$
9:    Update variable:          ▷ default step size as 1
$$\boldsymbol{z}^{(k+1)} = \boldsymbol{z}^{(k)} + \boldsymbol{p}^{(k)}$$
10:    Evaluate new residual: $\boldsymbol{g}^{(k+1)} = g(\boldsymbol{z}^{(k+1)})$
11:    Define differences:
$$\Delta\boldsymbol{z} = \boldsymbol{z}^{(k+1)} - \boldsymbol{z}^{(k)}, \quad \Delta\boldsymbol{g} = \boldsymbol{g}^{(k+1)} - \boldsymbol{g}^{(k)}$$
12:    Compute rank-1 vectors:
$$\boldsymbol{v}^{(k)} = (-I + \boldsymbol{U}\boldsymbol{V}^\top)\Delta\boldsymbol{z}$$
$$\boldsymbol{u}^{(k)} = \frac{\Delta\boldsymbol{z} - (-I + \boldsymbol{U}\boldsymbol{V}^\top)\Delta\boldsymbol{g}}{(\boldsymbol{v}^{(k)})^\top \Delta\boldsymbol{g}}$$
13:    Update low-rank factors:
$$\boldsymbol{U} \leftarrow [\boldsymbol{U}, \boldsymbol{u}^{(k)}], \quad \boldsymbol{V} \leftarrow [\boldsymbol{V}, \boldsymbol{v}^{(k)}]$$
14:    $k \leftarrow k + 1$
15: **end while**
16: **return** $z^* = \boldsymbol{z}^{(M)}$

---

**Anderson acceleration.** The Anderson solver accelerates convergence by updating the current state as a linear combination of previous iterates. The coefficients $\boldsymbol{\alpha}$ are obtained by solving a regularized optimization problem. To compute them, we employ a Lagrange multiplier formulation that yields a linear system, and fall back to a least-squares solution when the system becomes singular. The length of history states for interpolation is set to 5.

---

**Algorithm 5** Anderson acceleration

---

**Require:** Function $\varphi(\cdot, \boldsymbol{h})$, init $\boldsymbol{z}^{(0)}$, tol $\delta$, max iters $M$, history length $m$
**Ensure:** Equilibrium $\boldsymbol{z}^*$
 1: $\boldsymbol{z} \leftarrow \boldsymbol{z}^{(0)}$
 2: $k \leftarrow 0$
 3: **while** $k < M$ **do**
 4:     **if** $k < 2$ **then**
 5:         $\boldsymbol{z}^{(k+1)} \leftarrow \varphi(\boldsymbol{z}^{(k)}, \boldsymbol{h})$                                           ▷ no history yet
 6:     **else**
 7:         Let $m_k \leftarrow \min(m, k)$
 8:         Form difference matrices                                   ▷ $\in \mathbb{R}^{n \times m_k}$

$$G \leftarrow [\, \boldsymbol{z}^{(k-m_k+1)} - \boldsymbol{z}^{(k-m_k)}, \, \ldots, \, \boldsymbol{z}^{(k)} - \boldsymbol{z}^{(k-1)} \,]$$

 9:         Solve optimization problem:

$$\boldsymbol{\alpha} = \min_{\boldsymbol{\alpha} \in \mathbb{R}^{m_k}} \|G\boldsymbol{\alpha}\|_2^2, \text{ subject to } \mathbf{1}^T \boldsymbol{\alpha} = 1$$

10:         Leads to analytical form:

$$\begin{bmatrix} 0 & \mathbf{1}^\top \\ \mathbf{1} & G^\top G \end{bmatrix} \begin{bmatrix} \lambda \\ \boldsymbol{\alpha} \end{bmatrix} = \begin{bmatrix} \mathbf{1} \\ 0 \end{bmatrix}$$

11:         $\boldsymbol{z}' \leftarrow \sum_{i=1}^{m_k} \boldsymbol{\alpha}_i \boldsymbol{z}^{(k-i+1)}$                                        ▷ interpolation
12:         $\boldsymbol{z}^{(k+1)} \leftarrow \varphi(\boldsymbol{z}', \boldsymbol{h})$
13:     **end if**
14:     **if** $\|\boldsymbol{z}^{(k+1)} - \boldsymbol{z}^{(k)}\|_2 < \delta$ **then**
15:         **return** $\boldsymbol{z}^* = \boldsymbol{z}^{(k+1)}$
16:     **end if**
17:     $k \leftarrow k + 1$
18: **end while**
19: **return** $\boldsymbol{z}^* = \boldsymbol{z}^{(M)}$

---

### A.6 VISUALIZATION OF ADAPTIVE ITERATIONS

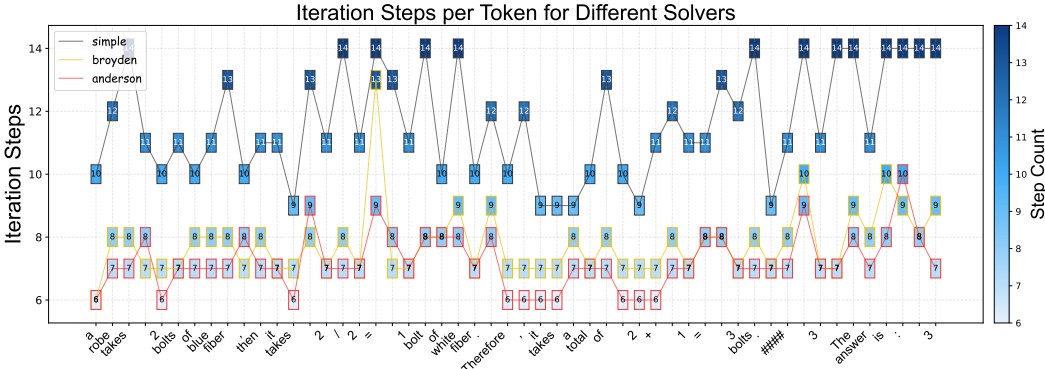

Figure 8: Iteration steps per token for different solver methods. Color intensity represents step count, with darker shades indicating more iterations required.

We visualize a generation example showing the number of iteration steps per token under the application of One-Step KV cache (Sec. 3.3) and fixed-point solvers (Sec. 3.4), using Qwen2.5-1.5B-Instruct as the base model. The pruning depth is set to $M = 14$, and the convergence threshold is $\delta = 0.5$. We represent the step counts as colored rectangles, where different colors indicate different numbers of iterations. As shown in Fig. 8, the solvers adaptively allocate different iteration steps across tokens, and the acceleration methods substantially reduce the overall computation compared to simple iteration. Consistent with the results in Table 4, Anderson acceleration achieves the best performance. In contrast, Broyden's method exhibits limited robustness, as its quasi-Newton update relies on low-rank Jacobian approximations without explicit step-size control, which often leads to unstable convergence or suboptimal fixed points in practice.

## A.7 EXPERIMENTAL DETAILS

The details of our experiments are elaborated as follows.

**Datasets and Prompts.** Our experiments are conducted on three downstream tasks.

*Commonsense reasoning.* The training dataset is composed of training sets from PIQA, SIQA, OBQA, MMLU and includes $154k$ instructions in total. The prompt is set as follows.

```
prompt="""
Choose the correct answer with the following question.\n
"""
```

*Mathematical solving.* The training dataset is from MetaMathQA, composed of $398k$ math instructions generated from the training sets from GSM8K and MATH. The prompt is set as follows and we adjust the output to produce final answers after 'The answer is:' in the end for convenience of extracting answers.

```
prompt="""
Please solve the following problem step by step. When you reach the
↪   answer, please output the answer after 'The answer is: ' at the
↪   end of the response.\n
"""
```

*Code generation.* The training dataset is from the educational_instruct and package_instruct part of OpenCoder's Stage 2 SFT dataset, composed of $289k$ Python coding instructions. The prompt is set as empty to adapt to the content of evaluation benchmarks HumanEval and MBPP.

The prompt template follows the default template of Qwen-2.5 & Llama-3.2 models in the library `transformers`. The content of templates are as follows. Variables `instruction` and `output` in the template denote specific content corresponding to each data entry.

```
Qwen

<|im_start|>system
You are a helpful assistant.<|im_end|>
<|im_start|>user
{prompt}
{instruction}<|im_end|>
<|im_start|>assistant
{output}<|im_end|>
```

```
Llama

<|begin_of_text|><|start_header_id|>system<|end_header_id|>

You are a helpful
↪   assistant.<|eot_id|><|start_header_id|>user<|end_header_id|>
```

```
{prompt}
{instruction}<|eot_id|><|start_header_id|>assistant<|end_header_id|>

{output}<|eot_id|>
```

**Baselines.** We reimplement baseline methods mentioned in Sec. 4 to align all the experimental conditions with our method.

*Dense* represents the finetuned original LLMs with LoRA for each downstream tasks.

*Sheared-LLaMA*(Xia et al., 2023) is a representative structural pruning method, which prunes attention heads, the intermediate channels of FFN, the hidden dimension and layers. It optimizes learnable $\ell_0$ regularization mask for subnetwork searching and applies LoRA to finetune the pruned models.

*Shortened-LLaMA*(Kim et al., 2024) utilizes Perplexity (PPL) to prune unimportant layers and apply LoRA to finetune the pruned models. The metric is evaluated based on the cross entropy:

$$\theta_{\text{PPL}}^l = \mathbb{E}_{x \sim \mathcal{C}} \exp\{-\frac{1}{T}\sum_t \log p_{\tilde{\mathcal{F}}_l}(x_t|x_{<t})\}, \tag{26}$$

where $\mathcal{C}$ represents the calibration dataset and $\tilde{\mathcal{F}}_l$ represents the transformer with layer $l$ removed. We pruned layers according to the descending order of $\theta_{\text{PPL}}^l$.

*ReplaceMe*(Shopkhoev et al., 2025) replaces intermediate consecutive transformer layers with a linear option. ReplaceMe(LS) utilizes least-square to solve the $\ell_2$-distance of feature alignment and ReplaceMe(Cosine) utilizes adam method to solve the cosine distance. It prunes the layer interval with cosine similarity and the first pruned layer index is defined as Eq. 15

*LLM-Streamline*(Chen et al., 2024) replaces intermediate consecutive transformer layers with a lightweight network. LLM-Streamline(Layer) utilizes a transformer Layer and LLM-Streamline(FFN) utilizes a feed-forward network. It aligns the feature with $\ell_2$-distance and prunes layers with cosine similarity (Eq. 15).

To clearly demonstrate the differences between baselines and our method, we take the experiments in mathematical solving task on Qwen2.5-1.5B-Instruct as an example and the implementation details are summarized in Table 9.

**Training.** For finetuning, we apply the consistent training settings across different baselines for fair comparisons. Specifically, we utilize Adam optimizer (Kingma & Ba, 2014) with a linear warmup on learning rate for the first 10% training epochs and cosine annealing schedule for the rest. The epochs of finetuning are shown in Table 9. The maximal learning rate is $2 \times 10^{-4}$ and the minimal is $2 \times 10^{-6}$. For hyper-parameters, we set $\lambda_{\text{distill}} = 1$ and linearly increase $\lambda_{\text{ce}}$ from 0 to 1 for the last 2 epochs. The context length of tokens is 2048. We set the clipping grad norm to 0.1 for stable training. The batch size on per device is 1. For pruning phase with training, our method and Sheared-LLaMA keep the same settings, which cost 2 epochs. GPPO mask is initialized as zero tensors and the size of reward set $p$ (the frequency of updating the reward model) is set to 10 and the frequency of updating the policy model $q$ is 2.

**Evaluation.** For commonsense reasoning tasks, we generate answers with greedy decoding. The answer is marked as correct if the model produces the same option with target in lower case. For mathematical solving tasks, we generate answers with greedy decoding. For GSM8K, the answer is marked as correct if the numerical difference with target is less than $10^{-5}$. For MATH, the answer is marked as correct if the answer string is the same as target. For code generation tasks, we generate 10 outputs with temperature as 0.9 and top_p as 0.9. We use bigcode-evaluation-harness(Ben Allal et al., 2022) to evaluate the pass@10 metric of outputs. To alleviate the misalignment in the format of generated code contents, we make simple post processing for benchmark HumanEval, that is preserving existing code contents (import necessary pacakges) in the prompt before concatenating the prompt and generation codes.

Table 9: Implementation details of baselines in mathematical solving task on Qwen2.5-1.5B-Instruct.

| | Dense | Sheared-LLaMA | Shortened-LLaMA | ReplaceMe | LLM-Streamline | ELM(Ours) |
|---|---|---|---|---|---|---|
| Pruning Metric/Modules | – | $\ell_0$ mask | Perplexity | Cosine Sim. | Cosine Sim. | GPPO mask |
| Pruning Calibration Data Size | – | FULL | $1k$ | $3k$ | $3k$ | FULL |
| Pruning Epochs | – | 2 | – | – | – | 2 |
| Pruning Layers | – | – | [2,3,4,5, 6,7,9,10] | [8,9,10,11, 12,13,14,15] | [8,9,10,11, 12,13,14,15] | [2,3,4,5, 6,7,8,9] |
| Finetuning Module | LoRA | LoRA | LoRA | Linear Transformation | Layer / FFN | Layer |
| Module Setting | $r = \alpha = 16$ | $r = \alpha = 64$ | $r = \alpha = 64$ | – | – | – |
| Parameter Size(M) | 18.5 | 52.8 | 52.8 | 2.25 | 46.8 / 41.3 | 51.5 |
| Finetuning Objective | SFT | SFT | SFT | Distill Loss (MSE / Cos) | Distill Loss (MSE) | Distill Loss (MSE) & SFT |
| Finetuning Data Size | FULL | FULL | FULL | $1k$ | FULL | FULL |
| Finetuning Epochs | 3 | 4 | 4 | 2 | 4 | 4 |

## A.8 GENERATION EXAMPLES

We sample several examples from mathematical solving and code synthesis to demonstrate the generation quality of ELM. The base model is Qwen2.5-1.5B-Instruct and the number of pruning layers $M = 8$.

---

### GSM8K

**Instruction:**
Gunter is trying to count the jelly beans in a jar. He asks his friends how many they think are in the jar. One says 80. Another says 20 more than half the first one. A third says 25% more than the first one. What is their average guess?

**ELM:**
The first friend says there are 80 jelly beans in the jar.
The second friend says there are 20 more than half of the first friend's guess, so 80/2 + 20 = 60 jelly beans.
The third friend says there are 25% more than the first friend's guess, so 80 + (80 * 0.25) = 80 + 20 = 100 jelly beans.
To find the average guess, we add up the guesses and divide by the number of guesses.
The total number of guesses is 3, so the average guess is (80 + 60 + 100) / 3 = 240 / 3 = 80 jelly beans.
#### 80
The answer is: 80

**Dense:**
The first friend says there are 80 jelly beans in the jar.
The second friend says there are 20 more than half of the first friend's guess, so 80/2 + 20 = 60 jelly beans.
The third friend says there are 25% more than the first friend's guess, so 80 + (80 * 0.25) = 80 + 20 = 100 jelly beans.
To find the average guess, we add up all the guesses and divide by the number of guesses.
The total number of jelly beans guessed is 80 + 60 + 100 = 240.
Since there are 3 guesses, the average guess is 240/3 = 80 jelly beans.
#### 80
The answer is: 80

---

### MATH

**Instruction:**
Given $\|\mathbf{v}\| = 5$ and $\|\mathbf{w}\| = 8$, find the largest possible value of

$$\|\operatorname{proj}_{\mathbf{w}} \mathbf{v}\|.$$

**ELM:**
The projection of $\mathbf{v}$ onto $\mathbf{w}$ is given by

$$\operatorname{proj}_{\mathbf{w}} \mathbf{v} = \frac{\mathbf{v} \cdot \mathbf{w}}{\|\mathbf{w}\|^2} \mathbf{w}.$$

The magnitude of this projection is

$$\|\operatorname{proj}_{\mathbf{w}} \mathbf{v}\| = \left\| \frac{\mathbf{v} \cdot \mathbf{w}}{\|\mathbf{w}\|^2} \mathbf{w} \right\| = \frac{|\mathbf{v} \cdot \mathbf{w}|}{\|\mathbf{w}\|^2} \|\mathbf{w}\|.$$

Since $\|\mathbf{w}\| = 8$, this becomes

$$\|\operatorname{proj}_{\mathbf{w}} \mathbf{v}\| = \frac{|\mathbf{v} \cdot \mathbf{w}|}{8^2} \cdot 8 = \frac{|\mathbf{v} \cdot \mathbf{w}|}{8}.$$

The largest possible value of $|\mathbf{v} \cdot \mathbf{w}|$ is $|\mathbf{v}| \cdot \|\mathbf{w}\| = 5 \cdot 8 = 40$, so the largest possible value of $\|\operatorname{proj}_{\mathbf{w}} \mathbf{v}\|$ is $\frac{40}{8} = \boxed{5}$.
The answer is: 5

**Dense:**
The projection of $\mathbf{v}$ onto $\mathbf{w}$ is given by

$$\operatorname{proj}_{\mathbf{w}} \mathbf{v} = \frac{\mathbf{v} \cdot \mathbf{w}}{\|\mathbf{w}\|^2} \mathbf{w}.$$

The magnitude of this projection is

$$\|\operatorname{proj}_{\mathbf{w}} \mathbf{v}\| = \left\| \frac{\mathbf{v} \cdot \mathbf{w}}{\|\mathbf{w}\|^2} \mathbf{w} \right\| = \frac{|\mathbf{v} \cdot \mathbf{w}|}{\|\mathbf{w}\|} = \frac{|\mathbf{v} \cdot \mathbf{w}|}{8}.$$

Since $\|\mathbf{v}\| = 5$ and $\|\mathbf{w}\| = 8$, the maximum value of $\mathbf{v} \cdot \mathbf{w}$ is $|\mathbf{v}|\|\mathbf{w}\| = 5 \cdot 8 = 40$.
Therefore, the largest possible value of $\|\operatorname{proj}_{\mathbf{w}} \mathbf{v}\|$ is $\frac{40}{8} = \boxed{5}$.
The answer is: 5

---

### HumanEval

**Instruction:**

```
def specialFilter(nums):
    """Write a function that takes an array of numbers as input and
     returns
    the number of elements in the array that are greater than 10
    and both
    first and last digits of a number are odd (1, 3, 5, 7, 9).
    For example:
    specialFilter([15, -73, 14, -15]) => 1
    specialFilter([33, -2, -3, 45, 21, 109]) => 2
    """
```

**ELM:**

```
def specialFilter(nums):
    def isSpecialNumber(n):
        str_n = str(n)
        return n > 10 and int(str_n[0]) % 2 != 0 and int(str_n[-1])
    % 2 != 0

    return sum(isSpecialNumber(x) for x in nums)
```

**Dense:**

```
def specialFilter(nums):
    count = 0
    for num in nums:
        if num > 10:
            str_num = str(num)
            if int(str_num[0]) % 2 != 0 and int(str_num[-1]) % 2 !=
 0:
                count += 1
    return count
```

### MBPP

**Instruction:**
Write a function to search an element in the given array by using binary search.
Tese case:
assert binary_search([1,2,3,5,8], 6) == False

**ELM:**

```
def binary_search(arr, x):
    low = 0
    high = len(arr) - 1
    mid = 0

    while low <= high:
        mid = (high + low) // 2

        if arr[mid] < x:
            low = mid + 1
        elif arr[mid] > x:
            high = mid - 1
        else:
            return True

    return False
```

**Dense:**

```
def binary_search(arr, x):
    low = 0
    high = len(arr) - 1
    mid = 0

    while low <= high:

        mid = (high + low) // 2

        if arr[mid] < x:
            low = mid + 1
        elif arr[mid] > x:
            high = mid - 1
        else:
            return True
    return False
```

## A.9    DISCUSSION ON PRUNING PATTERN

In the current paper, we focus on the simplest and most interpretable configuration: compressing a single continuous layer group into one fixed-point layer. This choice is aligned with prior observa-

tions that intermediate layers tend to be more compressible, and it allows us to clearly analyze the behavior of fixed-point compression without introducing additional degrees of freedom. However, more flexible pruning patterns, such as pruning multiple disjoint layer groups, are valuable extensions. To better understand this possibility, we conducted preliminary experiments and analysis on ELM and GPPO.

**Single Group vs. Multiple Groups.** We compare two pruning strategies under the same total pruning budget $M$=14: 1.prune one continuous group; 2.prune two disjoint groups (each with $M$=7).

To ensure fairness, we still use one fixed-point layer to compress all pruned layers. Experiments are conducted on coding tasks. Our preliminary results in Table 10 show that multi-group pruning performs worse than single-group pruning under the current ELM formulation. This is expected because when the pruned layers become disjoint, the fixed-point layer must jointly approximate multiple heterogeneous regions of the network, making the training significantly harder. This explains the consistent performance drop across both HumanEval and MBPP.

This observation also supports our choice of focusing on single continuous layer groups in the main paper, which is a stable and interpretable starting point for evaluating fixed-point compression.

Table 10: Different pruning patterns on ELM.

| $M$ | Pruned Layer Groups | HumanEval | MBPP |
|---|---|---|---|
| 14 | [2,3,4,5,6,7,8], [9,10,11,12,13,14,15] | 56.7 | 58.4 |
| 14 | [2,3,4,5,6,7,8], [10,11,12,13,14,15,16] | 47.0 | 57.0 |
| 14 | [2,3,4,5,6,7,8,9,10,11,12,13,14,15] | **58.5** | **65.4** |

**Extension of GPPO.** The GPPO algorithm in the paper is specifically designed for single-group pruning, consistent with our modeling focus. However, the framework is flexible enough to be extended. In response, we give a simple analysis of GPPO under multi-group pruning strategy. Considering pruning $M$ layers in total, we can list each pruning strategy as $[[l_1, M_1], \cdots, [l_k, M_k]]$, where $k$ is the total number of separate pruning layer groups, $l_i$ is the start layer index and $M_i$ is group size, subject to $l_i + M_i \leq l_{i+1}, l_k + M_k \leq N$. If there are $n$ valid strategies under the constraint, we can define a policy network $\theta \in \mathbb{R}^n$ producing a distribution over these $n$ candidates. LoRA adaptation can be applied per-strategy or per-layer depending on scale. In principle, GPPO can optimize the policy network $\theta$ similarly to the single-group case.

Enabling full arbitrary-pattern search requires substantially more design choices, such as how to parameterize multi-region compression and avoid combinatorial explosion. These go beyond the scope of this paper, but represent a promising direction for future exploration.

## A.10 THE USE OF LARGE LANGUAGE MODELS (LLMS)

LLMs were used solely as auxiliary tools for polishing the writing of this paper. Their role was limited to improving grammar, refining wording, and suggesting formatting adjustments for tables. No part of the research ideation, methodology, experiments, or analysis relied on LLMs. We understand that the authors bear full responsibility for all content of the paper, including any text generated or refined with the assistance of LLMs, and understand that such tools are not eligible for authorship.

## A.11 LICENSE OF ASSETS

We do not utilize any close-source assets in this paper. Llama3.2-3B-Instruct can be download from https://www.llama.com/llama-downloads/ and the license can be download from https://www.llama.com/llama3/license/. Qwen series model can be download from Huggingface and the license is Apache. All training datasets and evaluation benchmarks are obtained from Huggingface. The license of PIQA is Academic Free License v3. The license of SIQA, MBPP is CC BY-NC 4.0. The license of OBQA, MetaMathQA is Apache. And license for MMLU, GSM8K, MATH, OpenCoder, HumanEval is MIT acknowledge.

