# OpenReview forum: "Equilibrium Language Models"
_ICLR.cc/2026/Conference — ICLR 2026 Poster_

### Official Review · Reviewer_wdiM · 2025-10-30

**Soundness:** 3
**Presentation:** 3
**Contribution:** 3
**Rating:** 4
**Confidence:** 4

**Summary:**

This submission proposes Equilibrium Language Models (ELMs), a novel compression framework for Transformer-based large language models targeting edge deployment. The core idea is to replace groups of consecutive Transformer layers with a lightweight fixed-point network, reinterpreting deep computation as solving for the equilibrium state of a fixed-point system. To enable ELMs, the authors introduce two key components: (1) Group Pruning Policy Optimization (GPPO), a reinforcement learning-based method to automatically identify optimal layer pruning intervals; and (2) One-Step KVCache, which reduces memory overhead by storing only the final iteration’s cache without accuracy loss. Experiments on Qwen2.5 and Llama3.2 models show that ELMs prune 28% of parameters while retaining 99% of the accuracy of dense fine-tuned LLMs across commonsense reasoning, mathematical problem-solving, and code generation tasks.

**Strengths:**

1. The work breaks from traditional LLM pruning (e.g., weight-level, head-level, or heuristic layer pruning) by leveraging fixed-point network theory (from Deep Equilibrium Models, DEQs) to replace layer groups.

2. GPPO avoids the limitations of heuristic pruning metrics (cosine similarity, perplexity) by framing layer interval selection as a policy optimization problem.

3. One-Step KVCache is a critical optimization for edge deployment

**Weaknesses:**

1. All experiments use LLMs with exactly 28 layers (Qwen2.5-1.5B/7B, Llama3.2-3B). It remains unclear how ELMs perform on models with different depths.

2. The paper highlight the significance for edge deployment, however, the paper shows no data on latency results for edge devices.

3. While One-Step KVCache reduces memory, the submission only reports KV cache size (Table 3) but not total inference memory (including model parameters, activations). Edge devices often have strict RAM limits (e.g., 4GB on low-end phones), so total memory data is essential.

**Questions:**

1. What happens if the inherited Transformer layer is replaced with a simple FFN? Or if W_h is not initialized as an identity matrix? Such ablations would clarify which components drive performance retention.

2. How do you choose the convergence threshold for One-Step KVCache?

3. The paper mentions using Stochastic Jacobian-Free Backpropagation (SJFB) to reduce training cost , How does ELMs’ training time/GPU memory compare to baselines?

---

> ### Author Response · Authors · 2025-11-20
>
> We truly appreciate the reviewer for the valuable feedback. We have carefully considered the comments and suggestions and would like to address each of the concerns raised.
>
> **W1: ELMs on models with different depths.**
>
> We provide experiments over commonsense reasoning task and code synthesis based on Qwen2.5-3B-Instruct, which contains 36 layers and we prune 10 layers(27.8\%). The following table shows that ELM still achieves 98.3\% performance in average compared with Dense.
>
> || Ratio | PIQA | SIQA | OBQA | MMLU | HumanEval | MBPP | Avg. | RP |
> | - | - | - | - | - | - | - | - | - | - |
> | Dense | 0.00 | 85.7 | 80.7 | 87.0 | 61.0 | 87.8 | 75.8 | 79.7 | 100.0\% |
> | ELM | 27.8 | 85.3 | 80.5 | 84.8 | 60.5 | 86.0 | 73.0 | 78.4 | 98.3\% |
>
> **W2: Data on latency results.**
>
> We evaluate ELM with various solvers and thresholds on L20 GPU. Results show that with an average of only 1.1 iterations, ELM achieves 95\% of the original accuracy while decoding at 98.3 tokens/s, corresponding to a 28.7\% speedup over the Dense baseline. Acceleration solvers introduce negligible additional overhead, e.g., Anderson’s method still achieves a 14.9\% speedup with $\delta=4$, similar with simple solver.
>
> || threshold | decode speed(token/s) | iters | GSM8K Acc. |
> | - | - | - | - | - |
> | Dense | - | 76.4 | 8 | 72.4|
> | ELM(simple) | 0.0| 74.9 | 8 | 71.8|
> | ELM(broyden) | 4.0| 82.8 | 4.2 | 70.1|
> | ELM(anderson) | 4.0| 87.8 | 3.4 | 71.1|
> | ELM(simple) | 4.0| 87.2 | 3.6 | 71.6|
> | ELM(simple) | 10.0 | 93.6 | 2.4 | 70.1|
> | ELM(simple) | 100.0 | 98.3 | 1.1 | 68.8|
>
> **W3: Total inference memory (including model parameters, activations).**
>
> We reports memory efficiency of ELMs in the following table. ELM stores more 3KB activations for iteration hidden states, which is negligible during inference. The memory consumption of LLMs is primarily determined by model parameters and KV-cache. The baseline dense model occupies 3.09 GB, while our ELM model requires only 2.44 GB, yielding a 21\% reduction. Using One-Step KV-cache, ELM further reduces KV-cache memory to 86.0 MB at a 4k context length, achieving an additional 25\% savings compared to the dense baseline.
>
> | | model(GB) | KV-cache(MB) |
> | - | - | - |
> | Dense | 3.09 | 114.7 |
> | ELM | 2.44 | 86.0 |
>
> **Q1: Ablations on Transformer layer/FFN and $W_h$**
>
> - Fixed-point Module ablation
>
> We conduct ablation study on fixed-point layer module in commonsense reasoning task. ELM(FFN) behaves significant performance degeneration compared with ELM(Layer). The ablation indicates the necessity of using a transformer layer to compress the pruned module.
>
> || PIQA | SIQA | OBQA | MMLU |
> | - | - | - | - | - |
> | Layer | 83.1 | 76.9 | 83.8 | 54.0 |
> | FFN | 82.6 | 74.7 | 81.2 | 51.6 |
>
> - $W_h$ Initialization ablation
>
> We present the distillation loss value in the first 4k training steps on coding task. Results shows that initializing $W_h$ as identity significantly accelerates the convergence of ELM training compared with initializing parameters as random and also achieves better performance in MBPP benchmark.
>
> | $W_h$ | 0| 1k | 2k | 3k | 4k | Performance(MBPP) |
> | - | - | - | - | - | - | - |
> | identity | 2.172 | 0.272 | 0.198 | 0.152 | 0.154 | 70.6|
> | random | 210.000 | 0.824 | 0.459 | 0.398 | 0.355 | 69.8|
>
>
> **Q2: How do you choose the convergence threshold for One-Step KVCache?**
>
> We clarify that the threshold is used to terminate the iterative update when the residual between consecutive states falls below $\delta$(line 284), directly determines the number of iterative refinement steps. The One-Step KV Cache consistently stores the key–value from last iteration.
>
> Table 4 (line 437) in the paper provides an initial comparison of several threshold values. Here, we further extend the analysis with a wider range of thresholds using the simple solver. Results in table show that larger thresholds reduce the number of iterations and increase decoding speed, but may slightly degrade accuracy. In practical deployment, we find that:
> - $\delta=10$ offers a good balance: 97\% of dense model accuracy with 23\% decoding speedup.
> - For use cases requiring higher precision, a stricter threshold such as $\delta=4.0$ yields 99\% performance to baseline.
>
> || threshold | decode speed(token/s) | iters | GSM8K Acc. |
> | - | - | - | - | - |
> | Dense | - | 76.4 | 8 | 72.4|
> | ELM(simple) | 0.0| 74.9 | 8 | 71.8|
> | ELM(simple) | 4.0| 87.2 | 3.6 | 71.6|
> | ELM(simple) | 10.0 | 93.6 | 2.4 | 70.1|
> | ELM(simple) | 100.0 | 98.3 | 1.1 | 68.8|
>
> **Q3: ELMs’ training time/GPU memory compare to baselines(w/o SJFB)**
>
> We compare the training of ELM with or without SJFB based on Qwen2.5-1.5B-instruct in math task. The dataset size is 395k with 4 epochs. SJFB reduces about 14\% training time (1.0 GPU·days) and 13\% memory cost (2.6GB).
>
> || time(GPU·days) | memory(GB) |
> | - | - | - |
> | ELM w/o SJFB | 7.1| 20.1|
> | ELM with SJFB | 6.1| 17.5|

---

> ### Author Response · Authors · 2025-11-27
>
> Dear Reviewer,
>
> We have updated contents on inference efficiency of ELMs in global responses and we make additional supplementary in this response. As the discussion phase is drawing to a close, we kindly ask whether our responses have resolved your concerns or if there are any remaining issues that we can further clarify. Your insights are invaluable in refining our work, and we are eager to ensure that all your concerns are fully addressed.
>
> **W2: Data on latency results. \& Q2: How do you choose the convergence threshold for One-Step KVCache?**
>
> We expanded evaluation on ELMs' decoding speedups across thresholds, solvers, and pruning settings. For moderate thresholds (e.g., $\delta\in\{0.5,1.0,4.0\}$), ELM requires only 3–7 iterations and already achieves up to 15\% speedup over the Dense baseline, while maintaining 98\% of the original accuracy. Acceleration solvers produce similar accuracy and incur minimal overhead; Anderson’s method is often slightly faster under the same $\delta$. With larger thresholds ($\delta=10$), the iteration count drops to 2–3, yielding 28\% decoding speedup for $M=8$ and up to 50\% speedup for $M=14$, both maintain at least 95\% accuracy of small thresholds.
>
> For real deployment, simple and anderson solver are better choices for their stability and the threshold $\delta=10$ offers a good balance between decoding speedup and accuracy.
>
> | Method    | solver   | threshold | decode speed(token/s) | iters | GSM8K Acc. |
> |-----------|----------|-----------|-----------------------|-------|------------|
> | Dense     | --       | --        | 76.4                  | 8.0   | 72.4       |
> | ELM(M=8)  | simple   | 1.0       | 79.9                  | 5.7   | 70.6       |
> | ELM(M=8)  | simple   | 4.0       | 87.2                  | 3.6   | 71.6       |
> | ELM(M=8)  | simple   | 10.0      | 93.6                  | 2.4   | 70.1       |
> | ELM(M=8)  | simple   | 100.0     | 98.3                  | 1.1   | 68.8       |
> | ELM(M=8)  | broyden  | 1.0       | 80.5                  | 6.1   | 70.2       |
> | ELM(M=8)  | broyden  | 4.0       | 82.8                  | 4.2   | 70.1       |
> | ELM(M=8)  | broyden  | 10.0      | 92.8                  | 2.4   | 70.1       |
> | ELM(M=8)  | broyden  | 100.0     | 101.1                 | 1.1   | 68.9       |
> | ELM(M=8)  | anderson | 1.0       | 80.4                  | 4.8   | 70.8       |
> | ELM(M=8)  | anderson | 4.0       | 87.8                  | 3.4   | 71.1       |
> | ELM(M=8)  | anderson | 10.0      | 98.0                  | 2.4   | 70.4       |
> | ELM(M=8)  | anderson | 100.0     | 100.5                 | 1.1   | 68.8       |
> | ELM(M=14) | simple   | 1.0       | 89.3                  | 8.8   | 63.4       |
> | ELM(M=14) | simple   | 4.0       | 104.1                 | 5.5   | 63.8       |
> | ELM(M=14) | simple   | 10.0      | 115.6                 | 3.6   | 62.5       |
> | ELM(M=14) | simple   | 100.0     | 136.2                 | 1.1   | 35.8       |
> | ELM(M=14) | broyden  | 1.0       | 84.1                  | 6.5   | 63.6       |
> | ELM(M=14) | broyden  | 4.0       | 98.1                  | 4.6   | 61.0       |
> | ELM(M=14) | broyden  | 10.0      | 106.7                 | 3.5   | 59.0       |
> | ELM(M=14) | broyden  | 100.0     | 134.2                 | 1.1   | 36.2       |
> | ELM(M=14) | anderson | 1.0       | 91.3                  | 6.1   | 63.7       |
> | ELM(M=14) | anderson | 4.0       | 104.5                 | 4.3   | 62.6       |
> | ELM(M=14) | anderson | 10.0      | 115.3                 | 3.2   | 60.3       |
> | ELM(M=14) | anderson | 100.0     | 131.9                 | 1.1   | 35.5       |
>
>
> ---
> Thank you once again for your time and effort in reviewing our manuscript. We look forward to hearing from you and would greatly appreciate any further feedback.
>
> Best regards,
>
> The Authors of Submission 6608

---

### Official Review · Reviewer_Vx6z · 2025-10-31

**Soundness:** 3
**Presentation:** 3
**Contribution:** 3
**Rating:** 6
**Confidence:** 3

**Summary:**

This paper proposes Equilibrium Language Models (ELMs), a new family of Transformer-based language models designed to reduce inference-time memory usage and improve computational efficiency without sacrificing performance.

**Strengths:**

- Well-written and clearly structured; the mathematical formulations are concise and supported by clear intuition.

- Novel modeling approach: Reformulating decoding as a fixed-point equilibrium problem is original and elegant.

- Parameter efficiency: Demonstrates substantial parameter pruning without requiring fine-tuning.

**Weaknesses:**

- While the paper emphasizes parameter pruning (~28%), it does not report key deployment metrics such as inference latency, memory consumption, or speedup over dense baselines. This makes it difficult to verify the real-world efficiency gains, especially since equilibrium solvers may incur higher per-step computation even if they reduce parameter count.

- Although the paper presents an elegant formulation of Equilibrium Language Models (ELMs) and demonstrates strong pruning performance, it does not discuss the cost, stability, or convergence of obtaining these models. In particular, the Group Pruning Policy Optimization process likely introduces significant computational overhead, as it involves iterative search and training steps that may require multiple retraining cycles.

- In Table 2, ELMs are not consistently better than all baselines under the same compression level. For instance, LLM-Streamline (Layer) outperforms ELMs on MMLU at M=8, and Sheared LLM performs better on HumanEval. This suggests that ELMs may not be universally superior. A more systematic comparison across different  M values and tasks is necessary to understand when ELMs are advantageous or potentially weaker than existing methods.

**Questions:**

- What happens when the pruned layers are not contiguous? For example, instead of pruning 12 consecutive layers, what if we prune two separate groups of six layers distributed across the model?


I am open to discussing this further during the rebuttal and will be happy to increase my score if my concerns are addressed.

---

> ### Author Response · Authors · 2025-11-20
>
> We truly appreciate the reviewer for the valuable feedback. We have carefully considered the comments and suggestions and would like to address each of the concerns raised.
>
> **W1: Key deployment metrics such as inference latency, memory consumption, or speedup over dense baselines**
>
> We prodvide a comprehensive analysis about the practical efficiency of ELM on memory consumption and computation **in global response #1**.
>
> **W2: The cost, stability, or convergence of obtaining these models(In particular, GPPO).**
>
> We clarify the cost and stability for the training on ELM and GPPO based on Qwen2.5-1.5B-Instruct with $M=8$ in coding task.
>
> - ELM
>
> For the coding task with 289k dataset size, training a single ELM at 4 epochs requires 4.5 GPU·days with 17.5 GB memory. We present the distillation loss value in the first 4k training steps with ablations on the initialization of $W_h$ as identity or random matrix. With appropriate initialization, the training of ELM is stable and converges fast.
>
> | $W_h$ | 0 | 1k | 2k | 3k | 4k |
> | - | - | - | - | - | - |
> | identity | 2.172 | 0.272 | 0.198 | 0.152 | 0.154 |
> | random | 209.000 | 0.805 | 0.371 | 0.262 | 0.234 |
>
> - GPPO
>
> The cost of GPPO is modest and well justified by the performance gains it delivers. With 10k calibration dataset, training a single ELM requires 0.15 GPU·days with 17.5 GB memory usage, and enumerating all possible start layers would therefore require 0.15 × (28 − 8 + 1) = 3.15 GPU·days. GPPO employs a lightweight policy network and LoRA-based updates and completes in 0.2 hours (0.07 GPU·days) with 21.5 GB memory, introducing minimal extra overheads. We also illustrate the training loss curves of items in GPPO in supplementary A.2 (Fig.6, line 725). The figures shows that with 4k iteration steps on policy network the GPPO can produce the correct best option and the loss(rewards) converges quickly.
>
> | | Time(GPU·days) | Memory(GB) |
> | - | - | - |
> | ELM training | 0.15 | 17.5|
> | Enumeration | 3.15 | 17.5|
> | GPPO| 0.07 | 21.5|
>
> **W3: A more systematic comparison across different M values and tasks**
>
> We thank the reviewer for the constructive comment. We note that ELMs are already compared with layer-level pruning baselines under varying numbers of pruned layers in Table 2 (line 361). To further enhance the systematic comparison, we have expanded the evaluation to include Sheared-LLaMA and an additional benchmark (MMLU). The table below indicates that while ELMs outperforms existing pruning methods particularly with deeper compression. We will complete and include the full experimental settings in the final version.
>
> | Method| HumanEval | | | MBPP | | | MMLU | | |
> | - | - | - | - | - | - | - | - | - | - |
> | | M| | | M | | | M | | |
> | | 8| 10| 12| 8 | 10| 12| 8 | 10| 12|
> | Sheared-llama| **75.0** | 68.3 | 62.2 | 69.0 | 67.6 | 65.2 | 37.6 | 40.5 | 38.2 |
> | Shortened-llama| 64.0 | 59.1 | 53.0 | 65.6 | 61.8 | 60.2 | 41.5 | 39.4 | 36.9 |
> | LLM-Streamline(Layer) | 38.4 | 32.3 | 22.0 | 53.4 | 47.6 | 41.2 | 45.2 | 39.0 | 36.1 |
> | ELM | 74.4 | **71.3** | **66.5** | **70.0** | **69.4** | **67.4** | **54.0** | **53.0** | **50.2** |
>
> **Q1: Pruned layers are not contiguous?**
>
> In the current paper, we focus on one contiguous group configuration to clearly analyze the behavior of fixed-point compression without introducing additional degrees of freedom.
>
> However, we fully agree that more flexible pruning patterns, such as pruning multiple disjoint layer groups, are valuable extensions. To better understand this possibility, we conducted preliminary experiments and analysis to evaluate ELM under multi-group pruning. We compare two pruning strategies under the same total pruning budget $M=14$:
> 1. prune 1 continuous group
> 2. prune 2 disjoint groups(each with $M=7$)
>
> To ensure fairness, we still use one fixed-point layer to compress all pruned layers. Experiments are conducted on coding tasks.
>
> | M | Layer Groups | HumanEval | MBPP |
> | - | - | - | - |
> | 14 | [2,15] | 58.5 | 65.4 |
> | 14 | [2,8],[9,15] | 56.7 | 58.4 |
> | 14 | [2,8],[10,16] | 47.0 | 57.0 |
>
> Our preliminary results show that multi-group pruning performs worse than single-group pruning under the current ELM formulation. This is expected because when the pruned layers become disjoint, the fixed-point layer must jointly approximate multiple heterogeneous regions of the network, making the training significantly harder. This explains the consistent performance drop across both HumanEval and MBPP.
>
> Multi-group pruning is an interesting direction, while enabling it effectively requires additional modeling choices. We provide more detailed discussion in global response #3. We consider it a promising direction for future work and will include the clarifications in the revision.

---

> ### Author Response · Authors · 2025-11-27
>
> Dear Reviewer,
>
> We have updated contents on inference efficiency of ELMs in global responses and we make additional supplementary in this response on systematic comparison across different M values and tasks. As the discussion phase is drawing to a close, we kindly ask whether our responses have resolved your concerns or if there are any remaining issues that we can further clarify. Your insights are invaluable in refining our work, and we are eager to ensure that all your concerns are fully addressed.
>
> **W3: A more systematic comparison across different M values and tasks**
>
> We extend our study to multiple pruning budgets $M\in\{8,10,12,14\}$ and three representative benchmarks (HumanEval, MBPP, MMLU) compared with 3 methods(Sheared-llama, Shortened-llama, LLM-Streamline(Layer)). Across different base LLMs(Qwen2.5-1.5B-Instruct, Llama3.2-3B-Instruct) ELMs consistently deliver the strongest accuracy among pruning-based baselines, especially under high pruning ratio settings. Concretely, for Qwen2.5-1.5B, ELM degrades the slowest with larget pruning layers $M$, preserving up to 90–95\% of the dense performance on coding tasks even at $M=12$. ELM on Llama3.2-3B retains 50.3 MMLU at $M=14$, substantially higher than the 32.1–46.2 range of other methods.
>
> | Benchmark             | Method                | HumanEval |       |       | MBPP  |       |       | MMLU  |       |       |        |
> |-----------------------|-----------------------|-----------|-------|-------|-------|-------|-------|-------|-------|-------|--------|
> |                       |                       | M         |       |       | M     |       |       | M     |       |       |        |
> |                       |                       | 8         | 10    | 12    | 8     | 10    | 12    | 8     | 10    | 12    | 14     |
> | Qwen2.5-1.5B-Instruct | Sheared-llama         | **75.0**      | 68.3  | 62.2  | 69.0  | 67.6  | 65.2  | 37.6  | 40.5  | 38.2  | 32.1   |
> |                       | Shortened-llama       | 64.0      | 59.1  | 53.0  | 65.6  | 61.8  | 60.2  | 41.5  | 39.4  | 36.9  | 34.6   |
> |                       | LLM-Streamline(Layer) | 38.4      | 32.3  | 22.0  | 53.4  | 47.6  | 41.2  | 45.2  | 39.0  | 36.1  | 34.1   |
> |                       | ELM                   | 74.4      | **71.3**  | **66.5**  | **70.0**  | **69.4**  | **67.4**  | **54.0**  | **53.0**  | **50.2**  | **46.2**   |
> | Llama3.2-3B-Instruct  | Sheared-llama         | 69.0      | 67.5  | 66.9  | **69.8**  | 67.3  | 66.8  | 39.1  | 40.3  | 39.3  | 38.1   |
> |                       | Shortened-llama       | 73.2      | 67.7  | 64.6  | 68.6  | 69.6  | 66.0  | 49.7  | 50.1  | 49.5  | 46.2   |
> |                       | LLM-Streamline(Layer) | 76.2      | 75.0  | 23.8  | 67.0  | 64.0  | 42.0  | **50.8**  | **50.8**  | 50.1  | 32.1   |
> |                       | ELM                   | **76.8**      | **77.4**  | **73.2**  | **69.8**  | **71.0**  | **70.4**  | 50.5  | **50.8**  | **50.7**  | **50.3**   |
>
> ---
> Thank you once again for your time and effort in reviewing our manuscript. We look forward to hearing from you and would greatly appreciate any further feedback.
>
> Best regards,
>
> The Authors of Submission 6608

---

### Official Review · Reviewer_6rpi · 2025-11-01

**Soundness:** 2
**Presentation:** 3
**Contribution:** 2
**Rating:** 6
**Confidence:** 4

**Summary:**

This paper proposes ELMs, a model compression framework for LLMs. Instead of pruning individual weights or layers, ELMs replace groups of Transformer layers with a fixed-point network. This approach reinterpretes deep sequential computation as solving for an equilibrium state. The paper introduces: 1. Group Pruning Policy Optimization (GPPO): a reinforcement learning–based policy that learns optimal layer intervals for conversion to fixed-point modules; 2. One-Step KV Cache: a technique that stores only the final iteration’s cache to reduce memory cost during inference without sacrificing accuracy; 3. Adaptive Solvers (Broyden/Anderson acceleration): used to speed up convergence of fixed-point iterations.

Experiments on Qwen2.5-1.5B, Qwen2.5-7B, and Llama3.2-3B across commonsense reasoning, math, and code benchmarks show that ELMs prune ~28% of parameters while maintaining ≈99% of the dense model’s accuracy.

**Strengths:**

1. The paper reformulates deep computation as solving for an equilibrium state, which is quite novel.
2. The approach consistently outperforms all baselines, including recent strong pruning methods. The method still reaches good performance on hard reasoning tasks (e.g., GSM8K, MATH, HumanEval), where most compression methods fail.

**Weaknesses:**

1. The GRPO approach seems to only search for the start of the layers to be pruned under a fixed number of pruned layers. In this case, the search space is not very huge. How is the computation cost comparing to going through all of the possible start layers? Also, GPPO adds a non-trivial training loop with policy optimization, which might not be required for simpler pruning methods.
2. Evaluations are limited to post-finetuned LLMs. It remains unclear how ELMs generalize to instruction-tuned multi-task setups.

**Questions:**

1. How does the choice of fixed-point iteration solver (simple, Broyden, Anderson) affect latency in real-time inference on GPU/edge devices?
2. Can GPPO be applied jointly to multiple layer groups (non-contiguous intervals), or is it limited to a single interval per model?
3. What is the training cost of GPPO, and how is it comparing to enumerating method?

---

> ### Author Response · Authors · 2025-11-20
>
> We truly appreciate the reviewer for the valuable feedback. We have carefully considered the comments and would like to address each of the concerns raised.
>
> **W1 \& Q1: Computation cost of GPPO comparing to going through all of the possible start layers(enumerating method)? Also, GPPO adds a non-trivial training loop with policy optimization, which might not be required for simpler pruning methods.**
>
> We thank the reviewer for raising the question regarding the costs of GPPO. While GPPO does introduce an additional policy optimization stage, its overhead is modest and well justified by the performance benefits it provides.
>
> Concretely, for coding task with 10k calibration dataset, training a single ELM on Qwen2.5-1.5B-Instruct with $M=8$ layers pruned (starting from a fixed layer) at 4 epochs requires 0.44 hours on 8×H20 GPUs (0.15 GPU·days) with 17.5 GB memory usage. Enumerating all possible start layers would therefore require 0.15 × (28 − 8 + 1) = 3.15 GPU·days, which is clearly expensive. In contrast, GPPO employs a lightweight policy network and LoRA-based updates. With 2 epochs training, GPPO completes in 0.2 hours (0.07 GPU·days) with 21.5 GB memory, introducing minimal extra overheads.
>
> We also note that heuristic metrics (PPL, Cosine Similarity) require comparable computational time but fail to identify optimal pruning configurations, resulting in degraded performance. GPPO offers an automatic, efficient, and stable solution for selecting the optimal start layer of ELM.
>
> | | Time(GPU·days) | Memory(GB) | Start layer | Performance(MBPP) |
> | - | - | - | - | - |
> | ELM training | 0.15 | 17.5 | -- | -- |
> | Enumeration | 3.15 | 17.5 | 2 | 70.6 |
> | GPPO| 0.07 | 21.5 | 2 | 70.6 |
> | PPL | 0.99 | 10.1 | 8 | 66.6 |
> | Cos | 0.03 | 8.7 | 9 | 65.4 |
>
> **W2: Evaluations are limited to post-finetuned LLMs. It remains unclear how ELMs generalize to instruction-tuned multi-task setups.**
>
> We appreciate the reviewer’s comment on the generalization of ELMs. Our work mainly focuses on post-finetuning scenarios, where the model has already been adapted to a specific task distribution and efficient inference is the primary goal. This focus reflects practical use cases where users often deploy instruction-finetuned models for repeated, domain-specific workloads.
>
> We give a preliminary experiment with math \& code multi-task setting. We extract only 200k instructions from math and coding task and perform experiments on Qwen2.5-1.5B-Instruct (M=8). Results show that ELM still maintains 96\% performance of Dense. We plan to explore large-scale instruction-tuning to assess ELMs' generalization ability in future work.
>
> | | GSM8K | MATH | HumanEval | MBPP | Avg. |
> | - | - | - | - | - | - |
> | Dense | 69.9 | 32.1 | 73.2 | 68.2 | 60.9 |
> | ELM | 67.7 | 27.3 | 70.7 | 67.2 | 58.2 |
>
> **Q1: How does the choice of fixed-point iteration solver (simple, Broyden, Anderson) affect latency in real-time inference on GPU/edge devices?**
>
> We evaluate different fixed-point solvers based on Qwen2.5-1.5B-Instruct under $M=8$ in GSM8K. We conduct our inference on a single L20 GPU with precision BFloat16. Results show that with threshold $\delta=4.0$, Anderson solver achieves maximal decoding speed up. Broyden introduces extra latency as the convergence is slower with more iteration steps due to the imprecise approximation with rank-1 vectors.
>
> | | threshold | decode speed(token/s) | iters | GSM8K Acc. |
> | - | - | - | - | - |
> | Dense| -- | 76.4 | 8 | 72.4 |
> | ELM(broyden) | 4.0 | 82.8 | 4.2 | 70.1 |
> | ELM(anderson) | 4.0 | 87.8 | 3.4 | 71.1 |
> | ELM(simple) | 4.0 | 87.2 | 3.6 | 71.6 |
>
> **Q2: Can GPPO be applied jointly to multiple layer groups (non-contiguous intervals), or is it limited to a single interval per model?**
>
> Currently GPPO is specifically designed for single-group pruning, consistent with our modeling focus. However, the framework is flexible enough to be extended. In response we give a simple analysis of GPPO under multi-group pruning strategy. Considering pruning M layers in total, we can list each pruning strategy as $[[l_1,M_1],\cdots,[l_k,M_k]]$, where
> - $k$: the total number of separate pruning layer groups
> - $l_i$: start layer index
> - $M_i$: group size
>
> *s.t.* $l_i+M_i\leq l_{i+1}, l_k+M_k\leq N$
>
> If there are $n$ valid strategies under the constraint, we can define a policy network $\theta\in\mathbb{R}^{n}$ producing a distribution over these $n$ candidates. LoRA adaptation can be applied per-strategy or per-layer depending on scale. In principle, GPPO can optimize the policy network similarly to the single-group case.
>
> However, enabling full arbitrary-pattern search requires substantially more design choices, such as how to parameterize multi-region compression and avoid combinatorial explosion. These go beyond the scope of this paper, but represent a promising direction for future exploration.
>
> We provide comprehensive discussion on pruning pattern in global response #3 and we will include these contents in the revision.

---

> ### Author Response · Authors · 2025-11-27
>
> Dear Reviewer,
>
> We have updated contents on inference efficiency of ELMs in global responses and we make additional supplementary in this response on latency of different fixed-point solvers. As the discussion phase is drawing to a close, we kindly ask whether our responses have resolved your concerns or if there are any remaining issues that we can further clarify. Your insights are invaluable in refining our work, and we are eager to ensure that all your concerns are fully addressed.
>
> **Q1: How does the choice of fixed-point iteration solver (simple, Broyden, Anderson) affect latency in real-time inference on GPU/edge devices?**
>
> We conduct evaluation of different fixed-point solvers across various settings, including pruning layers $M=8,14$ and thresholds $\delta=4.0,10.0$. The following table shows that all solvers maintain stable accuracy in GSM8K and anderson solver achieves best acceleration, up tp 28\% on $M=8$. On $M=14$, simple demonstrates comparable performance with anderson. Broyden is unstable and the worst under most of the settings due to the low precision and additional costs of 1-rank approximation.
>
> | Method    | solver   | threshold | decode speed(token/s) | iters | GSM8K Acc. |
> |-----------|----------|-----------|-----------------------|-------|------------|
> | Dense     | --       | --        | 76.4                  | 8.0   | 72.4       |
> | ELM(M=8)  | simple   | 4.0       | 87.2                  | 3.6   | 71.6       |
> | ELM(M=8)  | broyden  | 4.0       | 82.8                  | 4.2   | 70.1       |
> | ELM(M=8)  | anderson | 4.0       | 87.8                  | 3.4   | 71.1       |
> | ELM(M=8)  | simple   | 10.0      | 93.6                  | 2.4   | 70.1       |
> | ELM(M=8)  | broyden  | 10.0      | 92.8                  | 2.4   | 70.1       |
> | ELM(M=8)  | anderson | 10.0      | 98.0                  | 2.4   | 70.4       |
> | ELM(M=14) | simple   | 4.0       | 104.1                 | 5.5   | 63.8       |
> | ELM(M=14) | broyden  | 4.0       | 98.1                  | 4.6   | 61.0       |
> | ELM(M=14) | anderson | 4.0       | 104.5                 | 4.3   | 62.6       |
> | ELM(M=14) | simple   | 10.0      | 115.6                 | 3.6   | 62.5       |
> | ELM(M=14) | broyden  | 10.0      | 106.7                 | 3.5   | 59.0       |
> | ELM(M=14) | anderson | 10.0      | 115.3                 | 3.2   | 60.3       |
>
> ---
> Thank you once again for your time and effort in reviewing our manuscript. We look forward to hearing from you and would greatly appreciate any further feedback.
>
> Best regards,
>
> The Authors of Submission 6608

---

### Official Review · Reviewer_4aYQ · 2025-11-01

**Soundness:** 3
**Presentation:** 4
**Contribution:** 3
**Rating:** 6
**Confidence:** 3

**Summary:**

This paper proposes a compression method for LLMs called "Equilibrium Language Models" (ELMs). The approach replaces a continuous block of layers in the model with a single lightweight equilibrium layer, which computes its output through iterative solving, thereby achieving parameter compression. To support this framework, the paper introduces three auxiliary components: 1) a reinforcement learning-based "Group Pruning Policy Optimization" (GPPO) method for automatically selecting the layer blocks to be replaced; 2) a "single-step KV cache" mechanism designed to reduce memory overhead during iterative inference; and 3) the application of existing numerical solvers to accelerate convergence. Experiments demonstrate that this method maintains performance close to the original model across multiple benchmarks after pruning approximately 28% of the parameters, outperforming several baseline approaches.

**Strengths:**

A New Perspective on Compression
The paper rethinks model pruning as an implicit depth problem. This offers a new way to approach this field, showing that we can use iterative computation to replace a large number of parameters.

Strong Empirical Results
The paper shows excellent results in balancing parameter reduction and accuracy, outperforming strong existing baselines. This demonstrates the great potential of the ELM framework.

Efficient Memory Optimization
The "One-Step KV Cache" is a simple but effective innovation. It directly solves the huge memory overhead that the iterative method could cause during LLM inference. This is especially valuable for long sequence generation.

**Weaknesses:**

Lack of Key Inference Latency Evaluation
The authors claim the method is friendly for edge deployment, but they don't provide any real-world speed data (like tokens/sec) to support this. The iterative process, the solver, and the convergence checks all add extra overhead. The final latency might even be worse than the original model. Without this data, the practical value of the paper is questionable.

Missing Analysis of GPPO's Practicality and Cost
Automatic search is nice, but its cost is critical to determine if it's feasible. The authors need to quantify the search cost of GPPO (e.g., how many GPU-days it takes) and discuss its cost relative to the final model training.

Limited Pruning Strategy
The method is limited to pruning a single, contiguous block of layers. This is a strong assumption and may not be the best approach. The authors should discuss why they made this choice (e.g., technical limitations or to simplify the problem) and explore the possibility of extending it to more flexible pruning patterns, like multiple, non-contiguous blocks.

**Questions:**

Same as weakness.

---

> ### Author Response · Authors · 2025-11-20
>
> We truly appreciate the reviewer for the valuable feedback. We have carefully considered the comments and suggestions and would like to address each of the concerns raised.
>
> **W1: Lack of Key Inference Latency**
>
> We are grateful for your comment and conduct evaluations focusing on runtime speed in realistic deployment settings. Experiments are performed on Qwen2.5-1.5B-Instruct with $M=8$ layers pruned in GSM8K. Inference was executed on a single L20 GPU with BFloat16 precision. Results show that with an average of only 1.1 iterations, ELM achieves 95\% of the original accuracy while decoding at 98.3 tokens/s, corresponding to a 28.7\% speedup over the Dense baseline. Acceleration solvers introduce negligible additional overhead, e.g., Anderson’s method still achieves a 14.9\% speedup with $\delta=4$, similar with simple solver.
>
> | | threshold | decode speed(token/s) | iters | GSM8K Acc. |
> | - | - | - | - | - |
> | Dense | -- | 76.4 | 8 | 72.4 |
> | ELM(simple) | 0.0 | 74.9 | 8 | 71.8 |
> | ELM(broyden) | 4.0 | 82.8 | 4.2 | 70.1 |
> | ELM(anderson) | 4.0 | 87.8 | 3.4 | 71.1 |
> | ELM(simple) | 4.0 | 87.2 | 3.6 | 71.6 |
> | ELM(simple) | 10.0 | 93.6 | 2.4 | 70.1 |
> | ELM(simple) | 100.0 | 98.3 | 1.1 | 68.8 |
>
> These results demonstrate that ELM provides practical improvements in decoding efficiency. Iteration count can be greatly reduced via acceleration methods(Broyden, Anderson) and computations in solvers introduce minimal extra latency. We provide further discussion on the efficiency of ELM in global response #1 and we will include these contents in the revision.
>
> **W2: Missing Analysis of GPPO's Practicality and Cost**
>
> The overheads of GPPO are lightweight and well justified by the performance benefits it provides. Concretely, for coding task with 10k calibration dataset, training a single ELM on Qwen2.5-1.5B-Instruct with $M=8$ layers pruned at 4 epochs requires 0.44 hours on 8×H20 GPUs (0.15 GPU·days) with 17.5 GB memory usage. Enumerating all possible start layers would therefore require 0.15 × (28 − 8 + 1) = 3.15 GPU·days, which is clearly expensive. In contrast, GPPO employs a lightweight policy network and LoRA-based updates. With 2 epochs training, GPPO completes in 0.2 hours (0.07 GPU·days) with 21.5 GB memory, introducing minimal extra overheads.
>
> | | Time(GPU·days) | Memory(GB) |
> | - | - | - |
> | ELM training | 0.15 | 17.5 |
> | Enumeration | 3.15 | 17.5 |
> | GPPO | 0.07 | 21.5 |
>
> We provide more detailed discussion on GPPO costs in global response #2 and we will include these contents in revision.
>
> **W3: Limited Pruning Strategy**
>
> In the current paper, we focus on the simplest and most interpretable configuration: compressing a single continuous layer group into one fixed-point layer. This choice is aligned with prior observations that intermediate layers tend to be more compressible, and it allows us to clearly analyze the behavior of fixed-point compression without introducing additional degrees of freedom.
>
> However, we fully agree that more flexible pruning patterns, such as pruning multiple disjoint layer groups, are valuable extensions. To better understand this possibility, we conducted preliminary experiments and analysis to evaluate ELM under multi-group pruning.
>
> We compare two pruning strategies under the same total pruning budget $M=14$:
> 1. prune 1 continuous group
> 2. prune 2 disjoint groups(each with $M=7$)
>
> To ensure fairness, we still use one fixed-point layer to compress all pruned layers. Experiments are conducted on coding tasks.
>
> | M | Layer Groups | HumanEval | MBPP |
> | - | - | - | - |
> | 14 | [2,15] | 58.5 | 65.4 |
> | 14 | [2,8],[9,15] | 56.7 | 58.4 |
> | 14 | [2,8],[10,16] | 47.0 | 57.0 |
>
> Results show that multi-group pruning performs worse than single-group pruning under the current ELM formulation. This is expected because when the pruned layers become disjoint, the fixed-point layer must jointly approximate multiple heterogeneous regions of the network, making the training significantly harder. This explains the consistent performance drop across both HumanEval and MBPP.
>
> This observation also supports our choice of focusing on single continuous layer groups in the main paper, which is a stable and interpretable starting point for evaluating fixed-point compression. We provide comprehensive discussion on the pruning pattern in global response #3 and we will include these contents in the revision.

---

> ### Author Response · Authors · 2025-11-27
>
> Dear Reviewer,
>
> We have updated contents on inference efficiency of ELMs in global responses and we make additional supplementary in this response. As the discussion phase is drawing to a close, we kindly ask whether our responses have resolved your concerns or if there are any remaining issues that we can further clarify. Your insights are invaluable in refining our work, and we are eager to ensure that all your concerns are fully addressed.
>
> **W1: Lack of Key Inference Latency**
>
> Our expanded evaluation demonstrates that ELM provides stable and configurable decoding speedups across thresholds, solvers, and pruning settings. For moderate thresholds (e.g., $\delta\in\{0.5,1.0,4.0\}$), ELM requires only 3–7 iterations and already achieves up to 15\% speedup over the Dense baseline, while maintaining 98\% of the original accuracy. Acceleration solvers produce similar accuracy and incur minimal overhead; Anderson’s method is often slightly faster under the same $\delta$.
>
> With larger thresholds ($\delta=10$), the iteration count drops to 2–3, yielding 28\% decoding speedup for M=8 and up to 50\% speedup for M=14. Even in these aggressive settings($\delta=100$), ELM still preserves usable performance for M=8, while M=14 introduces noticeable accuracy degradation; this confirms the expected trade-off.
>
> These results show that fixed-point decoding is not a bottleneck. Iteration counts remain small, solver overhead is negligible, and the accuracy–speed trade-off can be smoothly tuned by $\delta$.
>
> The tuples in the following table represent values corresponding to (simple, broyden, anderson).
> | Method    | threshold | decode speed(token/s) | iters            | GSM8K Acc.          |
> |-|-|-|-|-|
> | Dense     | --        | 76.4                  | 8.0              | 72.4                |
> | ELM(M=8)  | 0.5       | (77.3, 76.7, 77.3)    | (7.2, 7.2, 5.6)  | (71.3, 70.5, 71.0)  |
> | ELM(M=8)  | 1.0       | (79.9, 80.5, 80.4)    | (5.7, 6.1, 4.8)  | (70.6, 70.2, 70.8)  |
> | ELM(M=8)  | 4.0       | (87.2, 82.8, 87.8)    | (3.6, 4.2, 3.4)  | (71.6, 70.1, 71.1)  |
> | ELM(M=8)  | 10.0      | (93.6, 92.8, 98.0)    | (2.4, 2.4, 2.4)  | (70.1, 70.1, 70.4)  |
> | ELM(M=8)  | 100.0     | (98.3, 101.1, 100.5)  | (1.1, 1.1, 1.1)  | (68.8, 68.9, 68.8)  |
> | ELM(M=14) | 0.5       | (77.2, 75.4, 83.3)    | (12.3, 8.1, 7.5) | (63.8, 62.9, 64.3)  |
> | ELM(M=14) | 1.0       | (89.3, 84.1, 91.3)    | (8.8, 6.5, 6.1)  | (63.4, 63.6,63.7)   |
> | ELM(M=14) | 4.0       | (104.1, 98.1, 104.5)  | (5.5, 4.6, 4.3)  | (63.8, 61.0, 62.6)  |
> | ELM(M=14) | 10.0      | (115.6, 106.7, 115.3) | (3.6, 3.5, 3.2)  | (62.5, 59.0, 60.3)  |
> | ELM(M=14) | 100.0     | (136.2, 134.2, 131.9) | (1.1, 1.1, 1.1)  | (35.8, 36.2, 35.5)  |
>
> ---
> Thank you once again for your time and effort in reviewing our manuscript. We look forward to hearing from you and would greatly appreciate any further feedback.
>
> Best regards,
>
> The Authors of Submission 6608

---

### Author Response · Authors · 2025-11-20
**Global Response Part.1 (#1 Inference Efficiency)**

We sincerely thank the reviewers for their valuable feedback. We have carefully studied all comments, and these insightful suggestions are highly helpful for strengthening our work. Below, we summarize several common concerns raised across multiple reviews and provide global responses. All additional experiments will be further completed within the next two weeks and clarifications presented here will be integrated into the revised manuscript. This part of global response includes the inference efficiency of ELMs.

**#1 Inference Efficiency of ELM**

We appreciate the reviewers’ constructive suggestions regarding the practical efficiency of ELM on GPUs/edge devices. In response, we conducted further evaluations focusing on both memory footprint and runtime speed in realistic deployment. Experiments are performed on Qwen2.5-1.5B-Instruct with $M=8,14$ layers pruned in the benchmark GSM8K. Inference was executed on a single L20 GPU with BFloat16 precision.

**#1.1 Memory Efficiency**

During inference, the memory consumption of LLMs is primarily determined by model parameters and KV-cache size. The baseline dense model occupies 3.09 GB, while our ELM(M=8) requires only 2.44 GB, yielding a 21\% reduction in model memory and ELM(M=14) yields 39\% reduction. Using One-Step KV-cache, ELM(M=8) further reduces KV-cache memory to 86.0 MB at 4K context length, achieving 25\% savings compared to Dense and ELM(M=14) achieves 46\% savings.

|                      | M   | model(GB) | KV-cache(MB) |
| -------------------- | --- | --------- | ------------ |
| Dense                | --  | 3.09      | 114.7        |
| ELM(w/o One-Step KV) | 8   | 2.44      | 114.7        |
| ELM                  | 8   | 2.44      | 86.0         |
| ELM(w/o One-Step KV) | 14  | 1.88      | 114.7        |
| ELM                  | 14  | 1.88      | 61.6         |

**#1.2 Computational Efficiency**

Reviewers expressed concerns regarding whether the fixed-point iterations in ELM yield actual decoding speedups. We emphasize two key points:
- Iteration count can be greatly reduced via acceleration methods(Broyden, Anderson) and computations in solvers introduce minimal additional latency.
- SJFB training enables ELM to maintain high generation quality even with a relaxed convergence threshold $\delta$.

Our expanded evaluation demonstrates that ELM provides stable and configurable decoding speedups across thresholds, solvers, and pruning settings. For moderate thresholds (e.g., $\delta\in\{0.5,1.0,4.0\}$), ELM requires only 3–7 iterations and already achieves up to 15\% speedup over the Dense baseline, while maintaining 98\% of the original accuracy. Acceleration solvers produce similar accuracy and incur minimal overhead; Anderson’s method is often slightly faster under the same $\delta$.

With larger thresholds ($\delta=10$), the iteration count drops to 2–3, yielding 28\% decoding speedup for M=8 and up to 50\% speedup for M=14. Even in these aggressive settings($\delta=100$), ELM still preserves usable performance for M=8, while M=14 introduces noticeable accuracy degradation; this confirms the expected trade-off.

These results show that fixed-point decoding is not a bottleneck. Iteration counts remain small, solver overhead is negligible, and the accuracy–speed trade-off can be smoothly tuned by $\delta$.

The tuples in the following table represent values corresponding to (simple, broyden, anderson).
| Method    | threshold | decode speed(token/s) | iters            | GSM8K Acc.          |
|-|-|-|-|-|
| Dense     | --        | 76.4                  | 8.0              | 72.4                |
| ELM(M=8)  | 0.5       | (77.3, 76.7, 77.3)    | (7.2, 7.2, 5.6)  | (71.3, 70.5, 71.0)  |
| ELM(M=8)  | 1.0       | (79.9, 80.5, 80.4)    | (5.7, 6.1, 4.8)  | (70.6, 70.2, 70.8)  |
| ELM(M=8)  | 4.0       | (87.2, 82.8, 87.8)    | (3.6, 4.2, 3.4)  | (71.6, 70.1, 71.1)  |
| ELM(M=8)  | 10.0      | (93.6, 92.8, 98.0)    | (2.4, 2.4, 2.4)  | (70.1, 70.1, 70.4)  |
| ELM(M=8)  | 100.0     | (98.3, 101.1, 100.5)  | (1.1, 1.1, 1.1)  | (68.8, 68.9, 68.8)  |
| ELM(M=14) | 0.5       | (77.2, 75.4, 83.3)    | (12.3, 8.1, 7.5) | (63.8, 62.9, 64.3)  |
| ELM(M=14) | 1.0       | (89.3, 84.1, 91.3)    | (8.8, 6.5, 6.1)  | (63.4, 63.6,63.7)   |
| ELM(M=14) | 4.0       | (104.1, 98.1, 104.5)  | (5.5, 4.6, 4.3)  | (63.8, 61.0, 62.6)  |
| ELM(M=14) | 10.0      | (115.6, 106.7, 115.3) | (3.6, 3.5, 3.2)  | (62.5, 59.0, 60.3)  |
| ELM(M=14) | 100.0     | (136.2, 134.2, 131.9) | (1.1, 1.1, 1.1)  | (35.8, 36.2, 35.5)  |

These results collectively demonstrate that ELM provides substantial improvements in both memory consumption and decoding efficiency. We will incorporate these analyses in the revision. In future work, we plan to further investigate deployment on more constrained edge platforms such as mobile devices.

---

### Author Response · Authors · 2025-11-20
**Global Response Part.2 (#2 GPPO costs; #3 Pruning Pattern)**

This part of global response includes the costs of GPPO and discussion on pruning pattern.

**#2 Costs of GPPO**

We thank the reviewers for raising the question regarding the costs of GPPO. While GPPO does introduce an additional policy optimization stage, its overhead is modest and well justified by the performance benefits it provides.

Concretely, for coding task with 10k calibration dataset, training a single ELM on Qwen2.5-1.5B-Instruct with $M=8$ layers pruned (starting from a fixed layer) at 4 epochs requires 0.44 hours on 8×H20 GPUs (0.15 GPU·days) with 17.5 GB memory usage. Enumerating all possible start layers would therefore require 0.15 × (28 − 8 + 1) = 3.15 GPU·days, which is clearly expensive. In contrast, GPPO employs a lightweight policy network and LoRA-based updates. With 2 epochs of training, GPPO can converge to the optimal pruning strategy and the training takes 0.2 hours (0.07 GPU·days) with 21.5 GB memory, introducing minimal extra overheads.

We also note that heuristic metrics (PPL, Cosine Similarity) require comparable computational time but fail to identify optimal pruning configurations, resulting in degraded downstream performance. GPPO offers an automatic, efficient, and stable solution for selecting the optimal start layer in ELM construction. A comparison is summarized below and will be included in the revision.

| | Time(GPU·days) | Memory(GB) | Start layer | Performance(MBPP) |
| - | - | - | - | - |
| ELM training | 0.15| 17.5 | -- | -- |
| Enumeration | 3.15| 17.5 | 2| 70.6 |
| GPPO| 0.07| 21.5 | 2| 70.6 |
| PPL | 0.99| 10.1 | 8| 66.6 |
| Cos | 0.03| 8.7 | 9| 65.4 |

**#3 Pruning Pattern**

In the current paper, we focus on the simplest and most interpretable configuration: compressing a single continuous layer group into one fixed-point layer. This choice is aligned with prior observations that intermediate layers tend to be more compressible, and it allows us to clearly analyze the behavior of fixed-point compression without introducing additional degrees of freedom.

However, we fully agree that more flexible pruning patterns, such as pruning multiple disjoint layer groups, are valuable extensions. To better understand this possibility, we conducted preliminary experiments and analysis on ELM and GPPO.

**#3.1 Single Group vs. Multiple Groups**

We compare two pruning strategies under the same total pruning budget $M=14$:
1. prune 1 continuous group
2. prune 2 disjoint groups(each with $M=7$)

To ensure fairness, we still use one fixed-point layer to compress all pruned layers. Experiments are conducted on coding tasks.

| M | Layer Groups | HumanEval | MBPP |
| - | - | - | - |
| 14 | [2,15] | 58.5 | 65.4 |
| 14 | [2,8],[9,15] | 56.7 | 58.4 |
| 14 | [2,8],[10,16] | 47.0 | 57.0 |

Our preliminary results show that multi-group pruning performs worse than single-group pruning under the current ELM formulation. This is expected because when the pruned layers become disjoint, the fixed-point layer must jointly approximate multiple heterogeneous regions of the network, making the training significantly harder. This explains the consistent performance drop across both HumanEval and MBPP.

This observation also supports our choice of focusing on single continuous layer groups in the main paper, which is a stable and interpretable starting point for evaluating fixed-point compression.

**#3.2 Extension of GPPO**

The GPPO algorithm in the paper is specifically designed for single-group pruning, consistent with our modeling focus. However, the framework is flexible enough to be extended. In response, we give a simple analysis of GPPO under multi-group pruning strategy. Considering pruning $M$ layers in total, we can list each pruning strategy as $[[l_1,M_1],\cdots,[l_k,M_k]]$, where

- $k$: the total number of separate pruning layer groups
- $l_i$: start layer index
- $M_i$: group size

*s.t*. $l_i+M_i\leq l_{i+1}, l_k+M_k\leq N$.

If there are $n$ valid strategies under the constraint, we can define a policy network $\theta\in\mathbb{R}^{n}$ producing a distribution over these $n$ candidates. LoRA adaptation can be applied per-strategy or per-layer depending on scale. In principle, GPPO can optimize the policy network $\theta$ similarly to the single-group case.

However, enabling full arbitrary-pattern search requires substantially more design choices, such as how to parameterize multi-region compression and avoid combinatorial explosion. These go beyond the scope of this paper, but represent a promising direction for future exploration.

---
Overall, multi-group pruning is an interesting direction, while enabling it effectively requires additional modeling choices, e.g., how to parameterize multiple independent compression regions, how to align intermediate representations across disjoint spans, and how to manage the searching space in GPPO. We consider it a promising direction for future work and will include the clarifications in the revision.

---

### Author Response · Authors · 2025-12-01
**Summary for Area Chairs**

Dear Area Chairs,

Thank you very much for taking on the responsibility to manage our submission under the current emergency in the review process. We truly appreciate your time and effort you are dedicating to support the academic community.

---
Given the unusual situation, we would like to provide a clear summary of the status of our submission. We hope this will help you quickly and holistically evaluate our manuscript. The summarization includes two parts: 1. contributions and strengths of our work; 2. concerns and questions from reviewers and our responses \& revisions.

**Contributions and Strengths**

The positive recognition of our contribution can be summarized as follows:

1. We proposes a novel LLM compression method "Equilibrium Language Models" (ELMs) that replaces a continuous block of layers with a single lightweight equilibrium layer; (Reviewer `4aYQ`,`6rpi`,`Vx6z`,`wdiM`)

2. We propose "Group Pruning Policy Optimization" (GPPO), a reinforcement learning method for automatically selecting the layer groups to be replaced, outperforming heuristic pruning metrics; (Reviewer `wdiM`)

3. We propose "One-step KV cache" mechanism to reduce KV cache overhead, which is is a critical optimization for edge deployment; (Reviewer `4aYQ`,`wdiM`)

4. We deploy numerical solvers to accelerate convergence without additional training;

5. Experiments demonstrate that our method maintains 99\% performance close to the original model across multiple benchmarks and base LLMs after pruning approximately 28\% of the parameters, outperforming existed pruning baselines especially on hard reasoning tasks like math and coding. (Reviewer `4aYQ`,`6rpi`,`Vx6z`)

**Concerns and Revisions**

1. Realistic performance of inference efficiency. (Reviewer `4aYQ`,`6rpi`,`Vx6z`,`wdiM`)

We provide a comprehensive evaluation on memory consumption and decoding speedup to demonstrate the inference efficiency of our proposed ELM. And we give a detailed comparsion across different solvers and a wider range of thresholds. We response to each concerned reviewer and make a summary in **Global Response Part.1(#1)**.

The revision in our paper is in **section 4.6 (line 454)** and **table 3 (line 414), table 4 (line 439)**.

2. The training cost of GPPO. (Reviewer `4aYQ`,`6rpi`,`Vx6z`)

We provide a comprehensive evaluation on the costs of GPPO training to demonstrate the practicality of our proposed GPPO. The time, memory overhead and the final selected strategy demonstrates the advantages of GPPO. We response to each concerned reviewer and make a summary in **Global Response Part.2(#2)**.

The revision in our paper is in **appendix A.2 (line 756)**.

3. The different pruning pattern of ELM and GPPO adapted design. (Reviewer `4aYQ`,`6rpi`,`Vx6z`)

We provide a preliminary experiments on different pruning pattern of ELM, pruning layers as two disjoint groups. In addition, we give an anlaysis of GPPO algorithm design adpated to this pruning pattern. We response to each concerned reviewer and make a summary in **Global Response Part.2(#3)**.

The revision in our paper is in **appendix A.9 (line 1239)**.

4. A more systematic comparison across different M values and tasks. (Reviewer `Vx6z`)

We expand our experiments of comparing pruning baselines under varying pruning ratios in Table 2(line 362). We include Sheared-LLaMA and an additional benchmark (MMLU). The results show that ELM outperforms existing pruning methods particularly with deeper compression.

The revision in our paper is in **section 4.2 (line 362)**.

5. Additional ablations on ELM, including base LLMs with different depths, fixed-point module design, $W_h$ initialization, SJFB. (Reviewer `wdiM`)

We conduct our ablation experiments and the results demonstrate the effectiveness of our designs.

The revision in our paper is in **appendix A.3 (line 774)**.

We have updated the manuscript according to the reviewers’ valuable comments: all modifications are highlighted in **blue font** in the revised PDF. Unfortunately, our discussion did not receive a response from our reviewer before the rebuttal disruption. Nevertheless, we believe that the additional experiments and revisions we performed have substantially addressed the reviewers’ concerns and questions, which are valuable to improve our manuscript.

---

Thank you again for your time and effort during this special review. We deeply appreciate your dedication to preserve the integrity of the review process.

Sincerely,

The Authors of Submission 6608

---

### Meta-Review · Area_Chair_NaYg · 2026-01-08

**Summary:**

All reviewers acknowledge the idea of replacing a continuous block of layers with a single lightweight equilibrium layer for network pruning is novel. However, the reviewers pointed out quite a few major concerns, including missing the efficiency results in terms of memory cost, inference speed, and training cost, and missing detailed ablations such as different network depths and different numbers of replaced layers. The rebuttal addressed most of the concerns. The key highlight of “maintains 99% original performance while pruning approximately 28% of the parameters” is somewhat misleading. The gain in the inference speed is only about 14%. Performance, memory saving, and inference speed need to be placed together to give readers a comprehensive view, instead of listing them separately. Nevertheless, this work is a significant delta compared to the conventional network pruning works, with impressive results.

**Reviewer Concerns:**

See above.

**Reviewer Scores:**

The one giving the negative rating might upgrade the score.

---

### Decision · Program_Chairs · 2026-01-26

Accept (Poster)